# Beyond The Rainbow: High Performance Deep Reinforcement Learning on a Desktop PC

**Tyler Clark** [1]  **Mark Towers** [1]  **Christine Evers** [1]  **Jonathon Hare** [1]

## Abstract

Rainbow Deep Q-Network (DQN) demonstrated combining multiple independent enhancements could significantly boost a reinforcement learning (RL) agent's performance. In this paper, we present "Beyond The Rainbow" (BTR), a novel algorithm that integrates six improvements from across the RL literature to Rainbow DQN, establishing a new state-of-the-art for RL using a desktop PC, with a human-normalized interquartile mean (IQM) of 7.4 on Atari-60. Beyond Atari, we demonstrate BTR's capability to handle complex 3D games, successfully training agents to play Super Mario Galaxy, Mario Kart, and Mortal Kombat with minimal algorithmic changes. Designing BTR with computational efficiency in mind, agents can be trained using a high-end desktop PC on 200 million Atari frames within 12 hours. Additionally, we conduct detailed ablation studies of each component, analyzing the performance and impact using numerous measures. Code is available at https://github.com/VIPTankz/BTR.

## 1. Introduction

Deep Reinforcement Learning (RL) has achieved numerous successes in complex sequential decision-making tasks, most rapidly since Mnih et al. (2015) proposed Deep Q-Learning (DQN). With this success, RL has become increasingly popular among smaller research labs, the hobbyist community, and even the general public. However, recent state-of-the-art approaches (Schrittwieser et al., 2020; Badia et al., 2020a; Hessel et al., 2021; Kapturowski et al., 2023) are increasingly out of reach for those with more lim-

---
[1]School of Electronics and Computer Science, University Of Southampton, Southampton, UK. Correspondence to: Tyler Clark <tjc2g19@soton.ac.uk>.

*Proceedings of the $42^{nd}$ International Conference on Machine Learning*, Vancouver, Canada. PMLR 267, 2025. Copyright 2025 by the author(s).

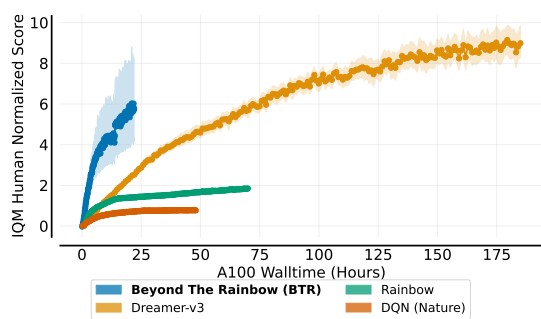

Figure 1: Interquartile mean human-normalized performance for BTR against other RL algorithms on the Atari benchmark in terms of walltime performance (all results use 200M frames). The results for DQN and Rainbow DQN are those reported in RLiable (Agarwal et al., 2021), and Dreamer-v3 refers to Hafner et al. (2023). Shaded areas show 95% bootstrapped confidence intervals, with BTR using 4 seeds.

ited compute resources, either in terms of the required hardware or the walltime necessary to train a single agent. This is a unique issue in RL compared to natural language processing or image recognition which have foundation models that can be efficiently fine-tuned for a new task or problem (Lv et al., 2024). Meanwhile, RL agents must be trained afresh for each environment. Therefore, the development of powerful RL algorithms that can be trained quickly on inexpensive hardware is crucial for smaller research labs and the hobbyist community.

These concerns are not new. Ceron & Castro (2021) highlighted that Rainbow DQN (Hessel et al., 2018) required 34,200 GPU hours (equivalent to 1435 days) of training, making the research impossible for anyone except a few research labs, with more recent algorithms exacerbating this problem. Recurrent network architectures (Horgan et al., 2018), high update to sample ratio (D'Oro et al., 2022), and the use of world-models and search-based techniques (Schrittwieser et al., 2020), all increase the computational resources necessary to train agents. Many of these use distributed approaches requiring multiple CPUs and GPUs (or TPUs), or requiring numerous days and weeks to train a single agent, dramatically decreasing RL's accessibility.

For this purpose, we develop "Beyond the Rainbow" (BTR), taking the same principle as Rainbow DQN (Hessel et al., 2018), selecting 6 previously independently evaluated improvements and combining them into a singular algorithm (Section 3). These components were chosen for their performance qualities or to reduce the computational requirements for training an agent. As a result, BTR sets a new state-of-the-art score for Atari-60 (Bellemare et al., 2013) (excluding recurrent approaches) with an Interquartile Mean (IQM) of 7.4[1] using a single desktop machine in less than 12 hours, and outperforms Rainbow DQN on Procgen (Cobbe et al., 2020) in less than a fifth of the wall-time (Section 4.1). Further, we demonstrate BTR's potential by training agents to solve three modern 3D games for the first time, Mario Kart Wii, Super Mario Galaxy and Mortal Kombat, that each contain complex mechanics and graphics (Section 4.2). To verify the effectiveness and effect of the six improvements to BTR, in Section 5.1, we conduct a thorough ablation of each component, plotting their impact on the Atari-5 environments and in Section 5.2, we utilize seven different measures to analyse the component's impact on the agent's policy and network weights. This allows us to more precisely understand how the components impact BTR beyond performance or walltime.

In summary, we make the following contributions to state-of-the-art RL.

- **High Performance (Section 4.1) -** BTR outperforms the state-of-the-art for non-recurrent RL on the Atari-60 benchmark, with an IQM of 7.4 (compared to Rainbow DQN's 1.9), outperforming humans on 52/60 games. Furthermore, BTR outperforms Rainbow DQN with Impala on the Procgen benchmark despite using a smaller model and 80% less walltime.

- **Modern Environments (Section 4.2) -** Testing beyond Atari, we demonstrate BTR can train agents for three modern games: Super Mario Galaxy (final stage), Mario Kart Wii (Rainbow Road), and Mortal Kombat (Endurance mode). These environments contain 3D graphics and complex physics and have never been solved using RL.

- **Computationally Accessible (Figure 6) -** Using a high-end desktop PC, BTR trains Atari agents for 200 million frames in under 12 hours, significantly faster than Rainbow DQN's 35 hours. This increases RL research's accessibility for smaller research labs and hobbyists without the need for GPU clusters or excessive walltime.

---

[1] All reported IQM scores use the best single evaluation for each environment throughout training as is standard, rather than the agent's score at 200 million, hence the discrepancy between the overall score and Figure 1.

- **Component Impact Analysis (Section 5) -** We conduct thorough ablations of BTR without each component, investigating performance and other measures. We discover that BTR widens action gaps (reducing the effects of approximation errors), is robust to observation noise, and reduces neuron dormancy and weight matrix norm (shown to improve plasticity throughout training).

## 2. Background

Before describing BTR's extensions, we outline standard RL mathematics, how DQN is implemented, and Rainbow DQN's extensions.

### 2.1. RL Problem Formulation

We adopt the standard formulation of RL (Sutton & Barto, 2018), described as a Markov Decision Process (MDP) defined by the tuple $(\mathcal{S}, \mathcal{A}, \mathcal{P}, \mathcal{R})$, where $\mathcal{S}$ is the set of states, $\mathcal{A}$ is the set of actions, $\mathcal{P} : \mathcal{S} \times \mathcal{A} \rightarrow \Delta(\mathcal{S})$ is the stochastic transition function, and $\mathcal{R} : \mathcal{S} \times \mathcal{A} \rightarrow \mathbb{R}$ is the reward function. The agent's objective is to learn a policy $\pi : S \rightarrow \Delta(\mathcal{A})$ that maximizes the expected sum of discounted rewards $\mathbb{E}_\pi[\sum_{t=0}^\infty \gamma^t r(s_t, a_t)]$, where $\gamma \in [0, 1)$ is the discount rate.

### 2.2. Deep Q-Learning (DQN)

One popular method for solving MDPs is Q-Learning (Watkins & Dayan, 1992) where an agent learns to predict the expected sum of discounted future rewards for a given state-action pair. To allow agents to generalize over states and thus be applied to problems with larger state spaces, Mnih et al. (2013) successfully combined Q-Learning with neural networks. To do this, training minimizes the error between the predictions from a parameterized network $Q_\theta$ and a target defined by

$$r_t + \gamma \max_{a \in A} Q_{\theta'}(s_{t+1}, a) , \tag{1}$$

where $Q_{\theta'}$ is an earlier version of the network referred to as the target network, which is periodically updated from the online network $Q_\theta$. The data used to perform updates is gathered by sampling from an Experience Replay Buffer (Lin, 1992), which stores states, actions, rewards, and next states experienced by the agent while interacting with the environment. To effectively explore the environment, $\epsilon$-greedy exploration is used, where each observation has a $\frac{\epsilon}{1}$ probability of choosing a random action.

### 2.3. Rainbow DQN and Improvements to DQN

In collecting 6 different improvements to DQN, Rainbow DQN (Hessel et al., 2018) proved cumulatively that these improvements could achieve a greater performance

than any individually. We briefly explain the individual improvements, ordered by performance impact, most of which are preserved within BTR (see Table 1). For more detail, we refer readers to the extension's respective papers:

1. **Prioritized Experience Replay -** To select training examples, DQN sampled uniformly from an Experience Replay Buffer, assuming that all examples are equally important to train with. Schaul et al. (2015) proposed sampling training examples proportionally to their last seen absolute temporal difference error, increasing training on samples for which the network most inaccurately predicts their future rewards.

2. **N-Step -** Q-learning utilizes bootstrapping to minimize the difference between the predicted value and the resultant reward plus the maximum value of the next state (Eq. 1). N-step (Sutton et al., 1998) reduces the reliance on this bootstrapped next value by considering the next $n$ rewards and the observation in $n$ timesteps (Rainbow DQN used $n = 3$).

3. **Distributional RL -** Due to the stochastic nature of RL environments and agent policies, Bellemare et al. (2017) proposed learning the return distribution rather than scalar expectation. This was done through modeling the return distributions using probability masses and the Kullbeck-Leibler divergence loss function.

4. **Noisy Networks -** Agents can often insufficiently explore their environment resulting in sub-optimal policies. Fortunato et al. (2018) added parametric noise to the network weights, causing the model's outputs to be randomly perturbed, increasing exploration during training, particularly for states where the agent has less confidence.

5. **Dueling DQN -** The agent's Q-value can be rewritten as the sum of state-value and advantage ($Q(s,a) = V(s) + A(s,a)$). Looking to improve action generalization, Wang et al. (2016) split the hidden layers into two separate streams for the value and advantage, recombining them with $Q(s,a) = V(s) + (A(s,a) - \frac{1}{|\mathcal{A}|}\sum_{a'} A(s,a'))$.

6. **Double DQN -** Selecting a target Q-value with the maximum Q-value from the next observation (Eq. 1) can frequently cause overestimation, negatively affecting agent performance. To reduce this overestimation, Van Hasselt et al. (2016) propose utilizing the online network rather than the target network to select the next action when forming targets, defined as:

$$r_t + \gamma Q_{\theta'}\left(s_{t+1}, \arg\max_{a \in A} Q_\theta(s_{t+1}, a)\right). \quad (2)$$

## 3. Beyond the Rainbow - Extensions and Improvements

Building on Rainbow DQN (Hessel et al., 2018), BTR includes 6 more improvements undiscovered in 2018.[2] Additionally, as hyperparameters are critical to agent performance, Section 3.2 discusses key hyperparameters and our choices. In the appendices, we include a table of hyperparameters, a figure of the network architecture and the agent's loss function (Appendices D.2, E and E.2). Finally, the source code using Gymnasium (Towers et al., 2024) is included within the supplementary material to help future work build upon or utilize BTR.

### 3.1. Extensions

**Impala Architecture + Adaptive Maxpooling** - Espeholt et al. (2018) proposed a convolutional residual neural network architecture based on He et al. (2016), featuring three residual blocks[3], substantially increasing performance over DQN's three-layer convolutional network. Following Cobbe et al. (2020), we scale the width of the convolutional layers by 2 to improve performance. We include an additional 6x6 adaptive max pooling layer after the convolutional layers (Schmidt & Schmied, 2021), which was found to speed up learning and support different input resolutions. The adaptive maxpooling is identical to a standard 2D maxpooling layer, but can be used with any input resolution as it automatically adjusts the stride and kernel size to fit a specified output size.

**Spectral Normalization (SN)** - To help stabilize the training of discriminators in Generative Adversarial Networks (GANs), Miyato et al. (2018) proposed Spectral Normalization to help control the Lipschitz constant of convolutional layers. SN works to normalize the weight matrices of each layer in the network by their largest singular value, ensuring that the transformation applied by the weights does not distort the input data excessively, which can lead to instability during training. Bjorck et al. (2021) and Gogianu et al. (2021) found that SN could improve performance in RL, especially for larger networks and Schmidt & Schmied (2021) found SN reduced the number of updates required before initial progress is made.

**Implicit Quantile Networks (IQN)** - Dabney et al. (2018) improved upon Bellemare et al. (2017), used in Rainbow DQN, learning the return distribution over the probability space rather than probability distribution over return val-

---

[2]After the completion of our work, we additionally found Layer Normalization applied after the stem of each residual block and between dense layers to be beneficial (see Appendix H for a discussion)

[3]The network architecture is referred to as Impala due to the accompanying training algorithm IMPALA proposed in Espeholt et al. (2018)

Table 1: A comparison of components between Rainbow DQN (Hessel et al., 2018) and BTR.

| Added To Rainbow DQN | Same As Rainbow DQN | Removed From Rainbow DQN |
|---|---|---|
| Impala (Scale=2) | N-Step TD Learning | Double (N/A with Munchausen) |
| Adaptive Maxpooling (6x6) | Prioritized Experience Replay | C51 (Upgraded to IQN) |
| Spectral Normalization | Dueling | |
| Implicit Quantile Networks | Noisy Networks | |
| Munchausen | | |
| Vectorized Environments | | |

ues. This removes the limit on the range of Q-values that can be expressed, and enables learning the expected return at every probability.

**Munchausen RL** - Bootstrapping is a core aspect of RL; used to calculate target values (Eq. 1) with most algorithms using the reward, $r_t$, and the optimal Q-value of the next state, $Q^*$. However, since in practice the optimal policy is not known, the current policy $\pi$ is used. Munchausen RL (Vieillard et al., 2020) leverages an additional estimate in the bootstrapping process by adding the scaled-log policy to the loss function (Eq. 3 where $\alpha \in [0, 1]$ is a scaling factor, $\sigma$ is the softmax function, and $\tau$ is the softmax temperature). This assumes a stochastic policy, therefore DQN is converted to Soft-DQN with $\pi_{\theta'} = \sigma(\frac{Q_{\theta'}}{\tau})$. As Munchausen does not use argmax over the next state, Double DQN is obsolete. Munchausen RL's update rule is

$$Q_\theta(s_t, a_t) = r_t + \alpha\tau \ln \pi_{\theta'}(a_t|s_t) +$$
$$\gamma \sum_{a' \in A} \pi_{\theta'}(a'|s_{t+1})(Q_{\theta'}(s_{t+1}, a') - \tau \ln(\pi_{\theta'}(a'|s_{t+1})) .$$
$$(3)$$

**Vectorization** - RL agents typically take multiple steps in a single environment, followed by a gradient update with a small batch size (Rainbow DQN took 4 environment steps, followed by a batch of 32). However, taking multiple steps in parallel and performing updates on larger batches can significantly reduce walltime. We follow Schmidt & Schmied (2021), taking 1 step in 64 parallel environments with one gradient update with batch size 256 (Schmidt & Schmied (2021) took two gradient updates). This results in a replay ratio (ratio of gradient updates to environment steps) of $\frac{1}{64}$. Higher replay ratios have been shown to improve performance (D'Oro et al., 2022), however we opt to keep this value low to reduce walltime.

### 3.2. Hyperparameters

Hyperparameters have repeatedly shown to have a very large impact on performance in RL (Ceron et al., 2024), thus we perform a small amount of tuning to improve per-

formance. Firstly, how frequently the target network is updated is closely intertwined with batch size and replay ratio. We found that updating the target network every 500 gradient steps[4] performed best. Given our high batch size, we additionally performed minor hyperparameter tests using different learning rates finding that a slightly higher learning rate of $1 \times 10^{-4}$ performed best, compared to $6.25 \times 10^{-5}$ in Rainbow DQN. In Appendix D.2, we clarify the meaning of the terms frames, steps and transitions.

For many years, RL algorithms have used a discount rate of 0.99, however, when reaching high performance, lower discount rates alter the optimal policy, causing even optimally performing agents to not collect the maximum cumulative rewards. To prevent this, we follow MuZero Reanalyse (Schrittwieser et al., 2021) using $\gamma = 0.997$. For our Prioritized Experience Replay, we use the lower value of $\alpha = 0.2$, the parameter used to determine sample priority, recommended by Toromanoff et al. (2019) when using IQN. Lastly, many previous experiments used only noisy networks or $\epsilon$-greedy exploration, however, we opt to use both until 100M frames, then set $\epsilon$ to zero, effectively disabling it. We elaborate on this decision in Appendix F.

## 4. Evaluation

To assess BTR, we test on two standard RL benchmarks, Atari (Bellemare et al., 2013) and Procgen (Cobbe et al., 2020) in Section 4.1. Secondly, we train agents for three modern games (Super Mario Galaxy, Mario Kart Wii, and Mortal Kombat) with complex 3D graphics and physics in Section 4.2, never before shown to be trainable with RL.

### 4.1. Atari and Procgen Performance

We evaluate BTR on the Atari-60 benchmark following (Machado et al., 2018) and without life information (see Appendix I for the impact), evaluating every million frames on 100 episodes. Figure 1 plots BTR against DQN, Rainbow DQN and Dreamer-v3, showing BTR's competitive performance despite using significantly less walltime. Fig-

---

[4]This equates to 32,000 environment steps (128,000 frames), compared to Rainbow DQN's 8,000 steps.

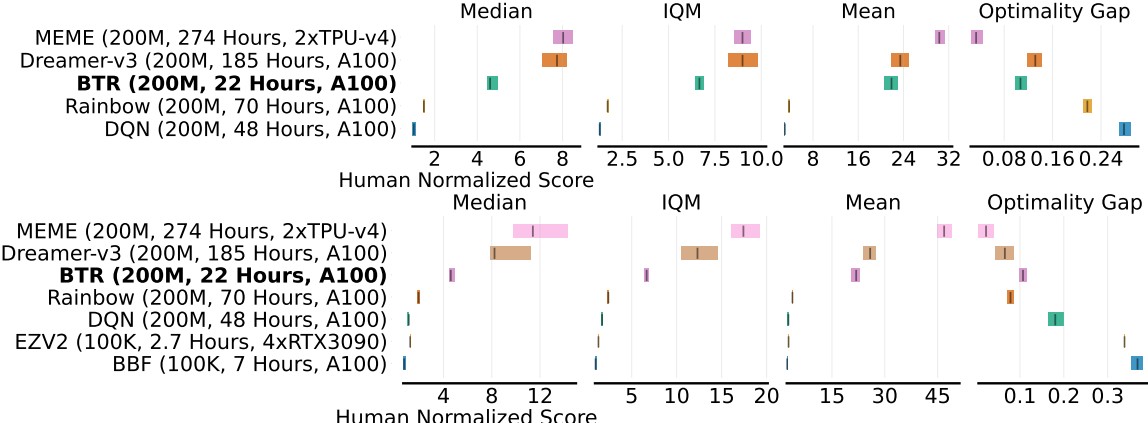

Figure 2: Box plot performance of BTR (4 seeds) against other popular algorithms such as MEME (Kapturowski et al., 2023), Dreamer v3 (Hafner et al., 2023), Bigger, Better, Faster (BBF) (Schwarzer et al., 2023) and EfficientZero-v2 (EZV2) (Wang et al., 2024). Brackets show the number of frames the algorithms use, the number of walltime hours and the hardware used respectively. Shaded areas show 95% confidence intervals. **Top:** Atari 55 game benchmark - we used the overlapping games 55 between the popular Atari-57 benchmark, and the 60 games used in RLiable (Agarwal et al., 2021). **Bottom:** Atari-26 benchmark, commonly used for testing sample-efficient algorithms.

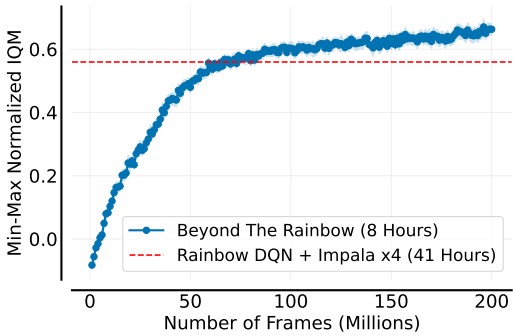

Figure 3: BTR compared to Rainbow DQN + Impala (width x4) (Cobbe et al., 2020) after 200M frames on the Procgen benchmark. Shaded areas show 95% CIs, with results averaged over 5 seeds.

ure 2 shows a box plot comparison of final performance. In comparison to human expert performance, BTR equals or exceeds them in 52 of 60. Importantly, we find that BTR appears to continue increasing performance beyond 200 million frames, indicating that higher performance is still possible with more time and data. Results tables and graphs can be found in Appendices A and B, respectively.

To further confirm BTR's performance, we benchmark on Procgen (Cobbe et al., 2020), a procedurally generated set of environments aiming to prevent overfitting to specific tasks, a prevalent problem in RL (Justesen et al., 2018; Juliani et al., 2019). The results are shown in Figure 3 with individual games in Appendix B. BTR is able to exceed Rainbow DQN + Impala's performance, despite using sig-

nificantly fewer convolutional filters (which Cobbe et al. (2020) found to significantly improve performance) and using 8 hours of walltime compared to 41. While BTR provides an improvement over Rainbow DQN in Procgen, we did not target procedurally generated environments thus it does not currently compete with the state-of-the-art (Cobbe et al., 2021; Hafner et al., 2023). There are numerous ways performance can be improved (Jesson & Jiang, 2024; Cobbe et al., 2020) which we leave to future work.

### 4.2. Applying BTR to Modern Games

To demonstrate BTR's capabilities beyond standard RL benchmarks, we utilized Dolphin (Dolphin-Emulator, 2024), a Nintendo Wii emulator, to train agents for a range of modern 3D games: Super Mario Galaxy, Mario Kart Wii and Mortal Kombat. Using a desktop PC, we were able to train the agent to complete some of the most difficult tasks within each game. Namely, the final level in Super Mario Galaxy, Rainbow Road (a notoriously difficult track in Mario Kart Wii), and defeating all opponents in Mortal Kombat Endurance mode. For details about the environments and setup, see Appendix J. To achieve this, BTR required minimal adjustments: changing the input image resolution to 140x114 (from Atari's 84x84) due to the game's higher resolution and aspect ratio, and to reduce the number of vectorized environments to 4 due to the games' memory and CPU requirements.

BTR was able to solve all three games, including consistently finishing first place in Mario Kart. In contrast, Rainbow DQN's performance plateaued before completing any of the games. We provide videos of our agent playing all

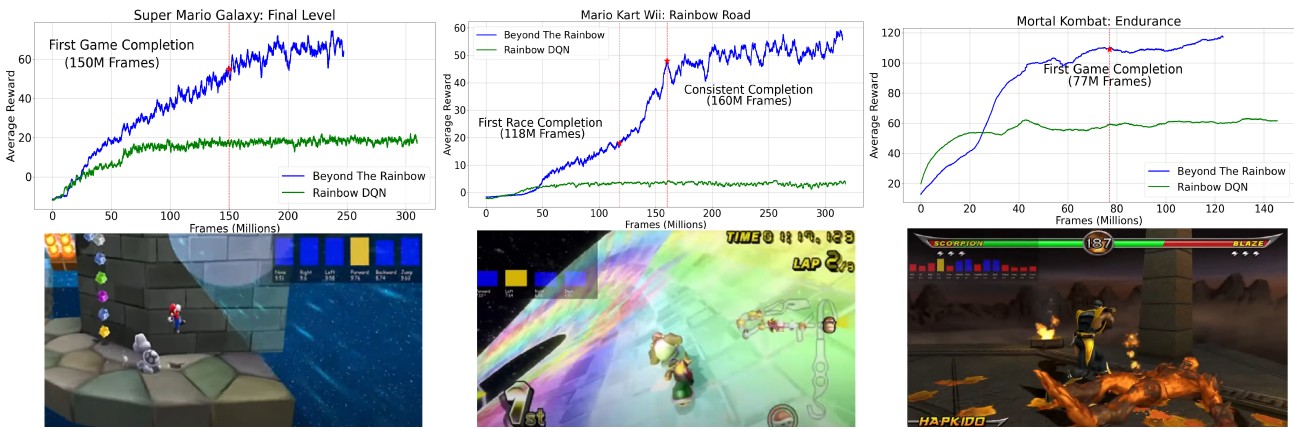

Figure 4: BTR being used to play Super Mario Galaxy (final level), Mario Kart Wii (Rainbow Road) and Mortal Kombat: Armageddon (Endurance Mode) respectively. Consistent completion is defined as over 90%.

three Wii games and all games in the Atari-5 benchmark [5].

## 5. Analysis

Given BTR's performance demonstrated in Section 4, in this section, we ablate each component to evaluate their performance impact (Section 5.1). Using the ablated agents, we measure numerous attributes during and after training to assess each component's impact (Section 5.2).

[5] https://www.youtube.com/playlist?list=PL4geUsKi0NN-sjbuZP_fU28AmAPQunLoI

### 5.1. Ablations Studies

BTR amalgamates independently evaluated components into a single algorithm. To understand and verify each component's contribution, Figure 5 plots BTR's performance without each component on the Atari-5 benchmark.[6]

We find that Impala had the largest effect on performance (+142% IQM), with the other components generally causing a less significant effect. Despite this, simply using Rainbow with Impala does not produce similar results (6.3 IQM compared to 7.7 on Atari-5). Munchausen and IQN have a strong impact on environments requiring fine-

[6] Due to the resources required to evaluate on all environments, Aitchison et al. (2023) proposes a subset of 5 games that closely correlate with the performance across all of them.

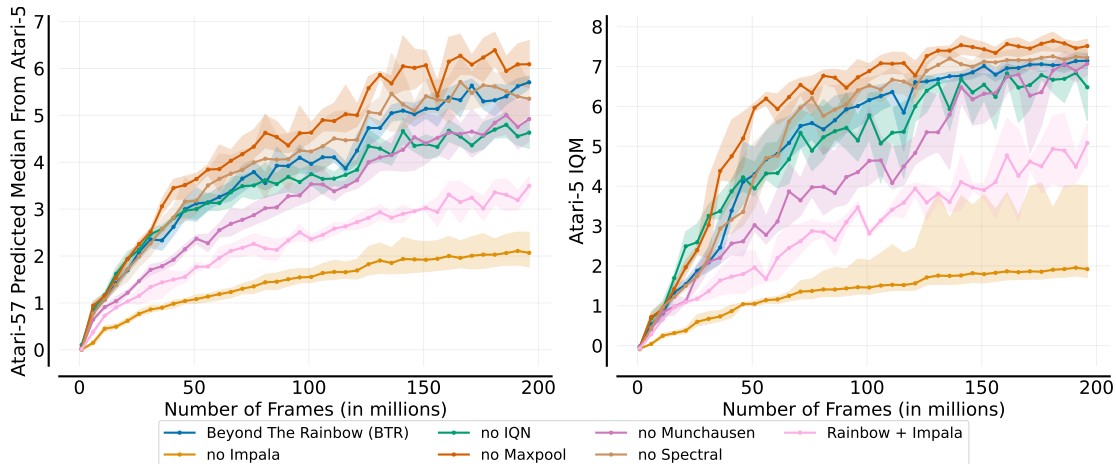

Figure 5: BTR's human-normalized scores without different components, with shaded areas showing 95% bootstrapped confidence intervals averaged over 4 seeds. **Left:** Predicted Atari-57 median score using the regression procedure defined in Aitchison et al. (2023). However, we find the prediction does not match the true median (see Appendix K). **Right:** Interquartile mean across the 5 games. For individual game graphs and additional ablations, see Appendices B and C.

grained control such as *Phoenix*, as explored in Section 5.2.

For vectorization and maxpooling, while their inclusion reduces performance, we find their secondary effects crucial to keep BTR computationally accessible. Omitting vectorization increases walltime by 328% (Figure 6) by processing environment steps in parallel and taking fewer gradient steps (781,000 compared to Rainbow DQN's 12.5 million).[7] We find that maxpooling decreases the model's parameters by 77%, and makes using wider convolutional layers possible without causing the total number of parameters to increase drastically.

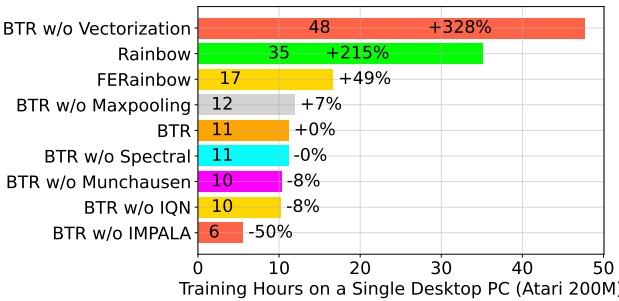

Figure 6: Walltime of BTR on a desktop PC with components removed, compared with Hessel et al. (2018) and Schmidt & Schmied (2021). For hardware details, see Appendix G.

### 5.2. What are the effects of BTR's components?

To help interpret the results in Section 5.1, Table 2 measures seven different attributes of the agent either during or after training: action gaps and action swaps (linked to causing approximation errors (Bellemare et al., 2016)); policy churn (which can cause excessive off-policyness and instability (Schaul et al., 2022)) and score with additional noise (indicating robustness of the policies).

While it is clear that Impala strongly contributes to performance, we find that without BTR's other components the learned policy is highly noisy and unstable. Table 2, demonstrates that without IQN and Munchausen the agent experiences very low action gaps (absolute Q-value difference between the highest two valued actions), causing the agent to swap its argmax action almost every other step. This is likely to result in approximation errors altering the policy and causing a high degree of off-policyness in the replay buffer. This is particularly detrimental in games requiring fine-grained control, such as *Phoenix* where the agent needs to narrowly dodge many projectiles, reflected in BTR's performance without these components.

---

[7]A result of removing vectorization is using smaller batches, which Obando Ceron et al. (2024) finds improves exploration.

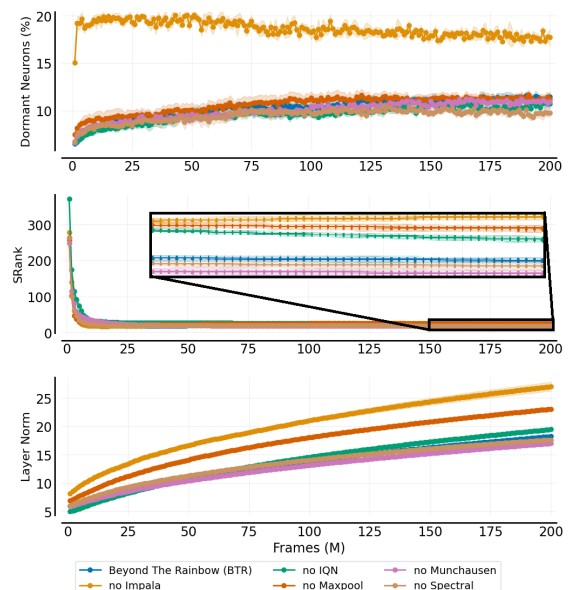

Figure 7: Plot showing % of dormant neurons (Sokar et al., 2023), SRank with $\delta = 0.01$ (Kumar et al., 2021) and L2 norm of network weights, for details see Appendix E.3. Results are averaged over 3 seeds and 5 tasks (Atari-5). Shading shows 95% confidence intervals.

Furthermore, we find that maxpooling produces a more robust policy. To test this, we evaluate the performance of BTR's ablations when taking different quantities of $\epsilon$-actions and with altered observations and find maxpooling alleviates some of the performance loss (Table 2). Lastly, we find Munchausen and IQN to have a significant impact on Policy Churn (Schaul et al., 2022), with Munchausen reducing it by 6.4% and IQN increasing it by 3.3%. As a result, when these components are used together, they appear to reach a level of churn which does not harm learning and potentially provides some exploratory benefits.

Lastly, Figure 7 shows an analysis of the trained model weights across the Atari-5 benchmark. We find little difference between trained models other than when removing Impala, which decreases dormant neurons and increases the L2 Norm of different layers, which have been linked with plasticity loss (Lyle et al., 2024).

## 6. Related Work

The most similar work to BTR, developing a computationally-limited non-distributed RL algorithm, is "Fast and Efficient Rainbow" (Schmidt & Schmied, 2021). They optimized Rainbow DQN to maximize performance for 10 million frames through parallelizing the environments and dropping C51 along with hyperparameter optimizations. This differs from our goals of producing

Table 2: Comparison of policy churn, action gaps, actions swaps and evaluation performance with different quantities of $\epsilon$-actions and color jitter (both only applied for evaluation). All measurements use the final agent, trained on 200 million frames, for Atari *Phoenix*, averaged over 3 seeds. Action Gap is the average absolute Q-value difference between the highest two valued actions. % Actions Swap is the percentage of times the agent's argmax action has changed from the last timestep. Policy churn is the percentage of states in which the agent's argmax action has changed after a single gradient step. Color jitter applies a random 10% change to the brightness, saturation and hue of each frame. For associated errors with these values, please see Appendix E.3.

| Category | BTR | w/o Munchausen | w/o IQN | w/o SN | w/o Impala | w/o Maxpool |
|---|---|---|---|---|---|---|
| Action Gap | 0.282 | 0.055 | 0.180 | 0.274 | 0.215 | 0.264 |
| % Action Swaps | 36.6% | 47.7% | 42.2% | 40.3% | 41.1% | 39.3% |
| Policy Churn | 3.8% | 11.0% | 0.5% | 3.3% | 4.5% | 4.2% |
| Score ColorJitter | **212k** | 85k | 110k | 187k | 19k | 187k |
| Score $\epsilon = 0.03$ | **94k** | 42k | 62k | 75k | 10k | 86k |
| Score $\epsilon = 0.01$ | **194k** | 70k | 110k | 132k | 13k | 171k |
| Score $\epsilon = 0$ | 330k | 184k | 187k | 296k | 21k | **406k** |

an algorithm that scales across training regimes (up to 200 million frames) and domains (Atari, Procgen, Super Mario Galaxy, Mario Kart and Mortal Kombat), resulting in different design decisions.

For less computation-limited approaches, Ape-X (Horgan et al., 2018) was the first to explore highly distributed training, allowing agents to be trained on a billion frames in 120 hours through using $> 100$ CPUs. Following this, Kapturowski et al. (2018) proposed R2D2 using a recurrent neural network, increasing sample efficiency but slowing down gradient updates by 38%. Agent57 (Badia et al., 2020a) was the first RL agent to achieve superhuman performance across 57 Atari games, though required 90 billion frames. MEME (Kapturowski et al., 2023), Agent57's successor, focused on achieving superhuman performance within the standard 200 million frames limit, achieved by using a significantly higher replay ratio and larger network architecture. Most recently, Dreamer-v3 (Hafner et al., 2023) used a 200 million parameter model requiring over a week of training, achieving similar results as MEME. We detail some key differences between BTR, MEME and Dreamer-v3 in Table 3. While these approaches perform equally or

better than BTR, all are inaccessible to smaller research labs or hobbyists due to their required computational resources and walltime. Therefore, while these algorithms have important research value demonstrating the possible performance of RL agents, performative algorithms with a lower cost of entry, like BTR, are necessary for RL to become widely applicable and accessible.

## 7. Conclusions

We have demonstrated that, once again, independent improvements from across Deep Reinforcement Learning can be combined into a single algorithm capable of pushing the state-of-the-art far beyond what any single improvement is capable of. Importantly, we find that this can be accomplished on desktop PCs, increasing the accessibility of RL for smaller research labs and hobbyists.

We acknowledge there exists many more promising improvements we could not include in BTR, leaving room for more future work to create stronger integrated agents in a few years. For example, BTR does not add an explicit exploration component, resulting in it struggling in hard-

Table 3: Comparison of performance, walltime, observations and complexity of different algorithms.

| Category | BTR | MEME | Dreamer-v3 |
|---|---|---|---|
| A100 GPU Days | 0.9 | Not Reported | 7.7 |
| Recurrent? | No (4 stacked frames) | Yes | Yes |
| Learns from? | Single Transitions | Trajectories (length 160) | Trajectories (length 64) |
| World Model | No | No | Yes |
| Parameters | 2.9M | Not Reported ($\approx >20$M) | 200M |
| Observation Shape | 84x84 | 210x160 | 64x64 |
| Gradient Steps | 781K | 3.75M | 1.5M |
| Atari-60 IQM | 7.4 | 9.6 | 9.6 |

exploration tasks such as *Montezuma's Revenge*; therefore, mechanisms used in Never Give Up (Badia et al., 2020b), etc may prove useful. Section 5.1 found that the neural network's core architecture, Impala, had the largest impact on performance, an area we believe is generally underappreciated in RL. Previous work (Kapturowski et al., 2018) has incorporated recurrent models enhancing performance, though we are uncertain how this can be incorporated into BTR without affecting its computational accessibility, a question which warrants future research.

## Impact Statement

This paper presents work whose goal is to advance the field of Reinforcement Learning, particularly to improve accessibility to those with limited computational resources. As with any work increasing accessibility, this has potential for misuse by bad actors. However, we believe these concerns are offset by the field's potential to tackle key societal problems.

## Acknowledgments

This work was supported by the UK Research and Innovation (UKRI) Centre for Doctoral Training in Machine Intelligence for Nano-electronic Devices and Systems [EP/S024298/1] and the Engineering and Physical Sciences Research Council (EPSRC) ActivATOR project [EP/W017466/1]. The authors acknowledge the use of the IRIDIS X High Performance Computing Facility, and the Southampton-Wolfson AI Research Machine (SWARM) GPU cluster generously funded by the Wolfson Foundation, together with the associated support services at the University of Southampton in the completion of this work. The authors dedicate this work to the memory of George Morton-Fallows, whose passion for computer science inspired this research.

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

## A. Full Results Tables

Table A1: Maximum scores obtained during training (averaged over 100 episodes and all performed using random seeds) after 200M Frames on the Atari-60 benchmark. Fast & Efficient Rainbow DQN and Munchausen-IQN refer to Schmidt & Schmied (2021) and (Vieillard et al., 2020) respectively. FE-Rainbow uses Life Information (See Appendix I), only 10M frames, and has missing games, so metrics are based on existing games.

| Game | Random | Human | DQN (Nature) | Rainbow | M-IQN | FE-Rainbow | BTR |
|---|---|---|---|---|---|---|---|
| AirRaid | 400 | 1000 | 7523 | 12472 | 19111 | | **51719** |
| Alien | 227 | 7127 | 2354 | 3610 | 4249 | 12508 | **18999** |
| Amidar | 5 | 1719 | 1268 | 2390 | 1653 | 2071 | **14027** |
| Assault | 222 | 742 | 1526 | 3490 | 6014 | 10709 | **19064** |
| Asterix | 210 | 8503 | 2803 | 16547 | 42615 | 346758 | **608829** |
| Asteroids | 719 | 47388 | 846 | 1494 | 1666 | 12345 | **153589** |
| Atlantis | 12850 | 29028 | 843372 | 791393 | 866810 | 812825 | **891773** |
| BankHeist | 14 | 753 | 560 | 1070 | 1305 | 1411 | **1482** |
| BattleZone | 2360 | 37187 | 18425 | 40316 | 50501 | 112652 | **168340** |
| BeamRider | 363 | 16926 | 5203 | 6084 | 12322 | 26398 | **110415** |
| Berzerk | 123 | 2630 | 467 | 832 | 719 | 3388 | **11417** |
| Bowling | 23 | 160 | 30 | 43 | 23 | 40 | **63** |
| Boxing | 0 | 12 | 79 | 98 | 99 | 99 | **100** |
| Breakout | 1 | 30 | 92 | 109 | 241 | 537 | **682** |
| Carnival | 380 | 4000 | 5111 | 4523 | 5588 | | **6284** |
| Centipede | 2090 | 12017 | 2378 | 6595 | 4425 | 8368 | **64242** |
| ChopperCommand | 811 | 7387 | 2722 | 13029 | 551 | 4208 | **956870** |
| CrazyClimber | 10780 | 35829 | 103549 | 146262 | **146419** | 140712 | 140927 |
| DemonAttack | 152 | 1971 | 5437 | 17411 | 63143 | 131657 | **135626** |
| DoubleDunk | -18 | -16 | -5 | 22 | 21 | -1 | **23** |
| ElevatorAction | 0 | 3000 | 408 | 79372 | **89237** | | 76941 |
| Enduro | 0 | 860 | 642 | 2165 | 2247 | 2266 | **2358** |
| FishingDerby | -91 | -38 | -1 | 42 | 54 | 42 | **58** |
| Freeway | 0 | 29 | 26 | 33 | 33 | **34** | 34 |
| Frostbite | 65 | 4334 | 482 | 8309 | 9419 | 5282 | **15158** |
| Gopher | 257 | 2412 | 5440 | 9987 | 23310 | 25606 | **97879** |
| Gravitar | 173 | 3351 | 209 | 1249 | 1105 | 2107 | **4253** |
| Hero | 1027 | 30826 | 15766 | 46290 | **25555** | 15377 | 25371 |
| IceHockey | -11 | 0 | -6 | 0 | 11 | 6 | **44** |
| Jamesbond | 29 | 302 | 671 | 995 | 1526 | | **59991** |
| JourneyEscape | -18000 | -1000 | -3300 | -1096 | -806 | | **2841** |
| Kangaroo | 52 | 3035 | 10744 | 13005 | 10704 | 11498 | **14300** |
| Krull | 1598 | 2665 | 6029 | 4368 | 10309 | 10324 | **11268** |
| KungFuMaster | 258 | 22736 | 22397 | 27066 | 25588 | 27444 | **53302** |
| MontezumaRevenge | 0 | 4753 | 0 | **500** | 0 | 0 | 0 |
| MsPacman | 307 | 6951 | 3431 | 3989 | 5630 | 5981 | **13200** |
| NameThisGame | 2292 | 8049 | 7549 | 8900 | 12440 | 19819 | **27917** |
| Phoenix | 761 | 7242 | 4993 | 8800 | 5315 | 60954 | **427481** |
| Pitfall | -229 | 6463 | -45 | -27 | -32 | -1 | **0** |
| Pong | -20 | 14 | 16 | 20 | 19 | **21** | 21 |
| Pooyan | 500 | 1000 | 3452 | 4344 | 13096 | | **22003** |
| PrivateEye | 24 | 69571 | 1113 | **21353** | 100 | 253 | 100 |
| Qbert | 163 | 13455 | 9801 | 18332 | 13159 | 25712 | **42927** |
| Riverraid | 1338 | 17118 | 9725 | 20675 | 16143 | | **25192** |
| RoadRunner | 11 | 7845 | 38430 | 55104 | 60370 | 81831 | **579800** |
| Robotank | 2 | 11 | 59 | 67 | 71 | 70 | **83** |
| Seaquest | 68 | 42054 | 2416 | 9590 | 23885 | 63724 | **428263** |
| Skiing | -17098 | -4336 | -16281 | -29268 | -10404 | -22076 | **-7938** |
| Solaris | 1236 | 12326 | 1478 | 1686 | 1835 | 2877 | **6301** |
| SpaceInvaders | 148 | 1668 | 1797 | 4455 | 10810 | 28098 | **54262** |
| StarGunner | 664 | 10250 | 48498 | 57255 | 64875 | 310403 | **577547** |
| Tennis | -23 | -8 | -3 | 0 | 0 | 15 | **24** |
| TimePilot | 3568 | 5229 | 3704 | 11959 | 14600 | 31333 | **113801** |
| Tutankham | 11 | 167 | 103 | 244 | 205 | 167 | **297** |
| UpNDown | 533 | 11693 | 8797 | 37936 | 197043 | | **391439** |
| Venture | 0 | 1187 | 13 | **1537** | 978 | 437 | 1 |
| VideoPinball | 0 | 17667 | 38720 | 460245 | 508012 | 269619 | **573774** |
| WizardOfWor | 563 | 4756 | 1473 | 7952 | 11352 | 15518 | **47314** |
| YarsRevenge | 3092 | 54576 | 23963 | 46456 | 106929 | 98908 | **209499** |
| Zaxxon | 32 | 9173 | 4471 | 14983 | 14286 | 18832 | **52619** |
| IQM (↑) | 0.000 | 1.000 | 0.771 | 1.852 | 2.181 | ≈ 2.769 | **7.361** |
| Median (↑) | 0.000 | 1.000 | 0.731 | 1.506 | 1.559 | ≈ 1.906 | **4.690** |
| Mean (↑) | 0.000 | 1.000 | 2.261 | 4.152 | 5.260 | ≈ 7.700 | **21.574** |
| Optimality Gap (↓) | 0.000 | 1.000 | 0.407 | 0.200 | 0.224 | ≈ 0.180 | **0.098** |
| Best | - | - | 0 | 3 | 3 | 2 | **54** |
| >Human | - | - | 22 | 43 | 34 | 38 | **52** |
| Surround | 7 | -10 | | | | | **10** |
| Defender | 2875 | 18689 | | | | 169929 | **461380** |

Table A2: Maximum scores obtained during training (averaged over 100 episodes and all performed 3 random seeds) after 200M Frames on the Atari-5 Environment, compared against other non-recurrent non-distributed algorithms. FE-Rainbow refers to Fast and Efficient Rainbow DQN (Schmidt & Schmied, 2021), and M-IQN refers to Munchausen-IQN (Vieillard et al., 2020). Metrics do not use the recommended regression procedure, as explained in Appendix K.

| Game | Random | Human | Rainbow DQN (Dopamine) | Rainbow DQN (Full) | M-IQN | FE-Rainbow | BTR |
|---|---|---|---|---|---|---|---|
| BattleZone | 2360 | 37188 | 40895 | 62010 | 52517 | 112652 | **168340** |
| DoubleDunk | -19 | -16 | 22 | 0 | 22 | -1 | **23** |
| NameThisGame | 2292 | 8049 | 9229 | 13136 | 12761 | 19819 | **27917** |
| Phoenix | 761 | 7243 | 8605 | 108529 | 5327 | 60955 | **427481** |
| QBert | 164 | 13455 | 18503 | 33818 | 14739 | 25712 | **42927** |
| IQM | 0.000 | 1.000 | 1.265 | 3.583 | 1.452 | 4.070 | **7.739** |
| Median | 0.000 | 1.000 | 1.21 | 2.532 | 1.44 | 3.167 | **4.766** |
| Mean | 0.000 | 1.000 | 3.714 | 5.817 | 3.745 | 4.684 | **18.453** |

Table A3: Comparison of performance and walltime against PQN (Gallici et al., 2024). PQN only reports results at 400M frames and includes life information, which greatly affects performance (see Appendix I). To provide a fairer comparison, we also report our results using life information but only use 200M frames. Below are Atari-5 IQM and per-game Scores, with BTR averaged over 3 seeds. Human-Normalized scores are reported for individual games, with the raw score in brackets.

| Game | BTR (with life info, 200M frames) | PQN (with life info, **400M** frames) |
|---|---|---|
| Inter-Quartile Mean | **12.18** | 3.86 |
| BattleZone | **12.73 (445,827)** | 1.51 (54,791) |
| DoubleDunk | **14 (23.0)** | 6.03 (-0.92) |
| NameThisGame | **4.51 (28,834)** | 3.18 (20,603) |
| Phoenix | **90.85 (589,662)** | 38.79 (252,173) |
| QBert | **9.79 (130,348)** | 2.37 (31,716) |
| Walltime (A100) | 22 Hours | **2 Hours** |
| Backend | (PyTorch (non-compiled) + gymnasium async) | (JAX + envpool) |

## B. Full Results Graphs

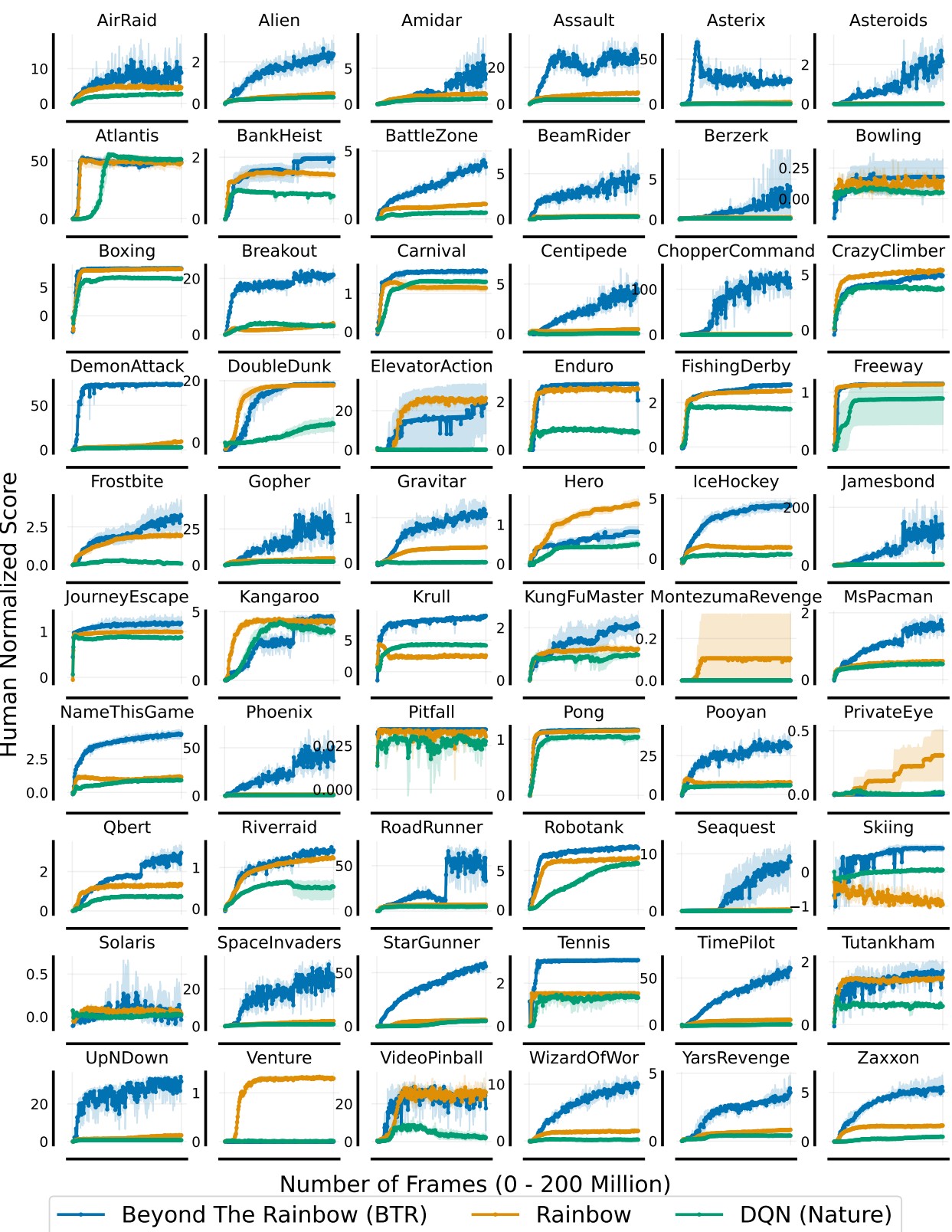

Figure B1: Performance of BTR (4 Seeds) on each individual game in all 60 Atari games. Shaded areas show 95% confidence intervals over different seeds.

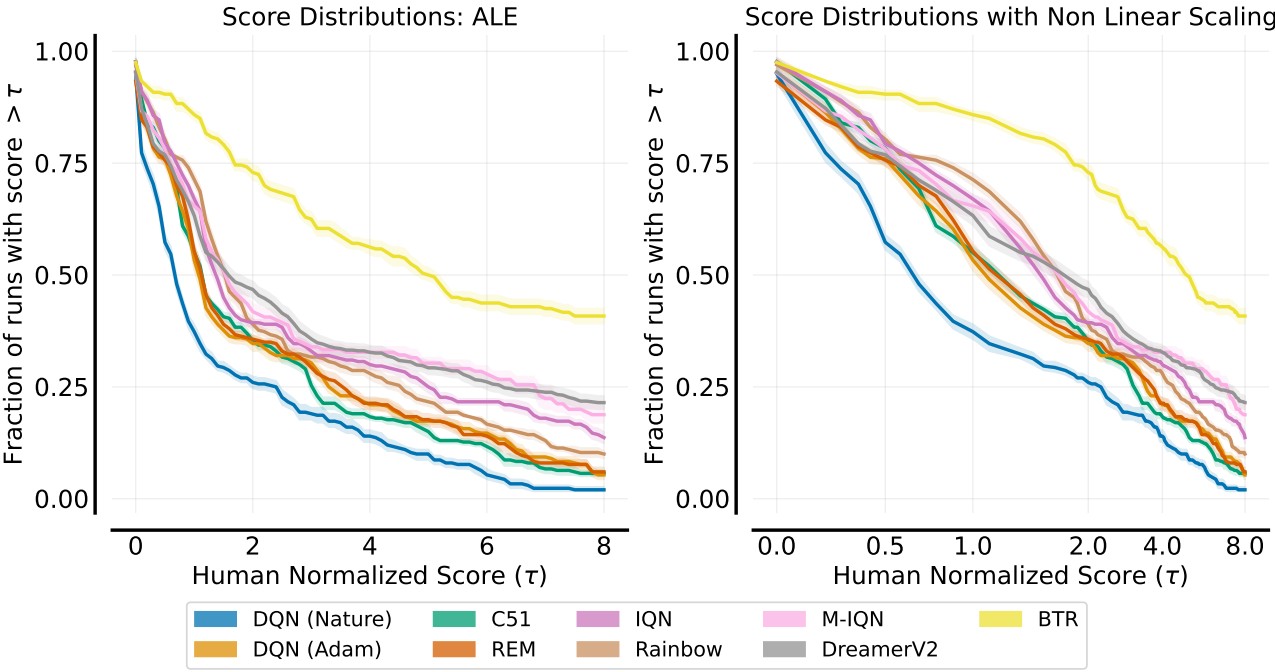

Figure B2: Final performance of BTR on Atari-60 (as used in RLiable (Agarwal et al., 2021)), against other popular algorithms. The plot displays performance profiles, with 95% confidence intervals and 4 seeds for BTR.

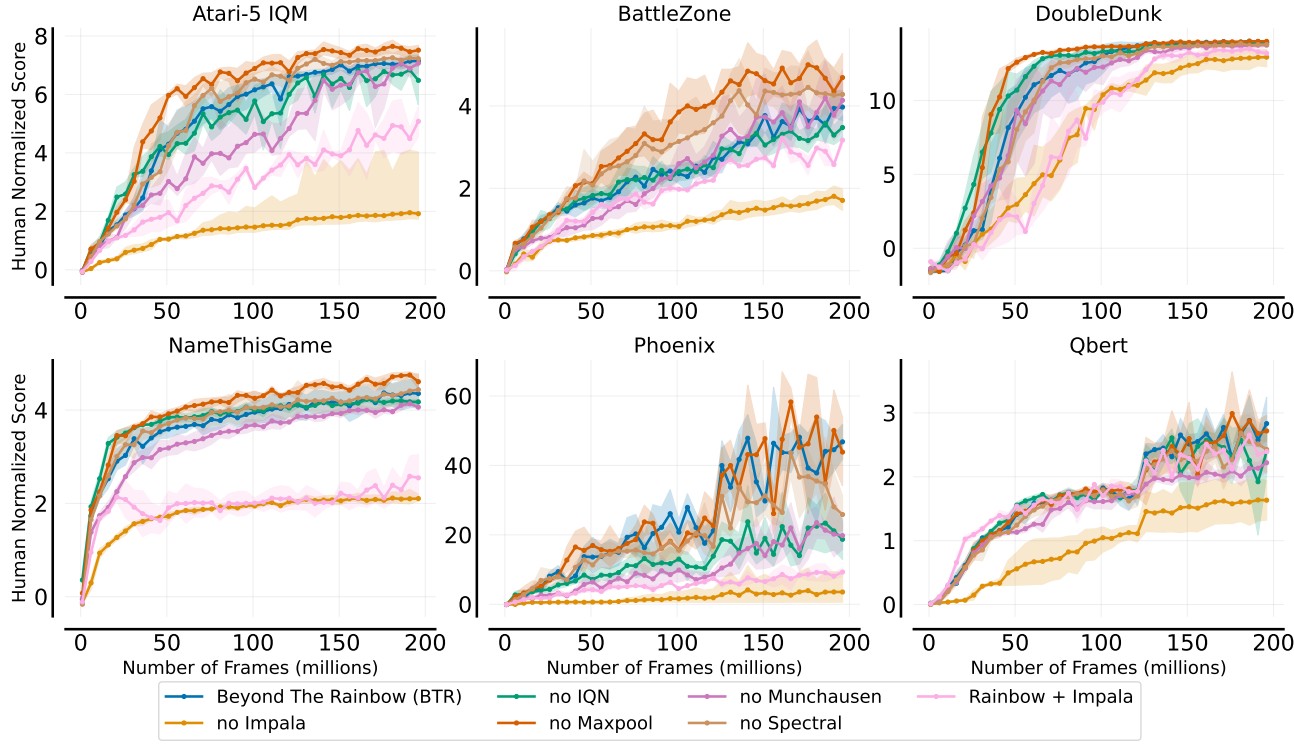

Figure B3: Performance of BTR with different components removed on individual games, and interquartile mean in the top left. Results are averaged over 4 seeds, with shaded areas showing 95% bootstrapped confidence intervals.

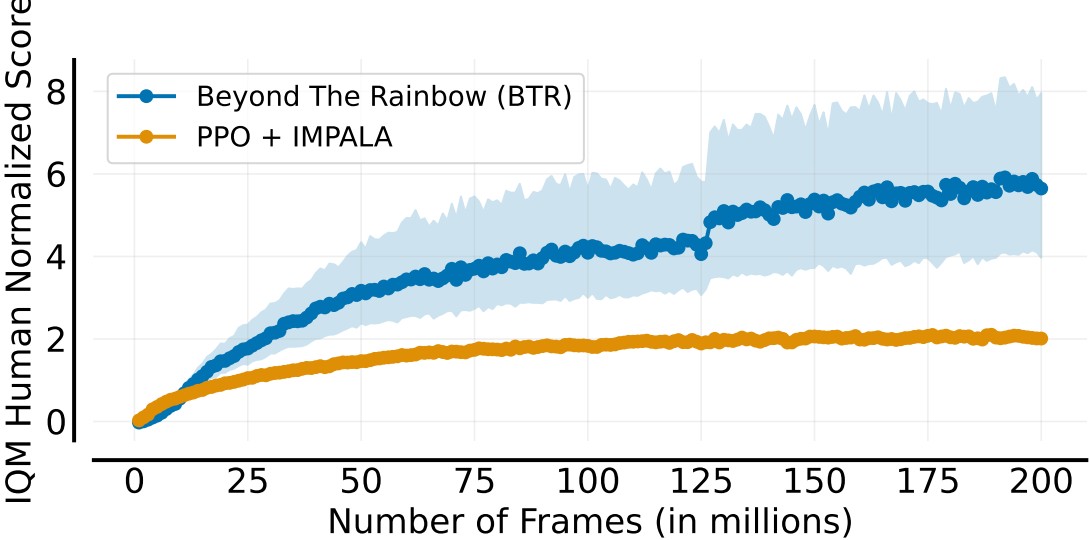

Figure B4: Figure shows BTR against PPO on Atari 57 (PPO's Atari 60 scores have not been reported). PPO uses the results provided by DreamerV3 (Hafner et al., 2023), which additionally uses vectorization and the impala algorithm. Shaded areas show 95% confidence intervals with BTR using 4 seeds.

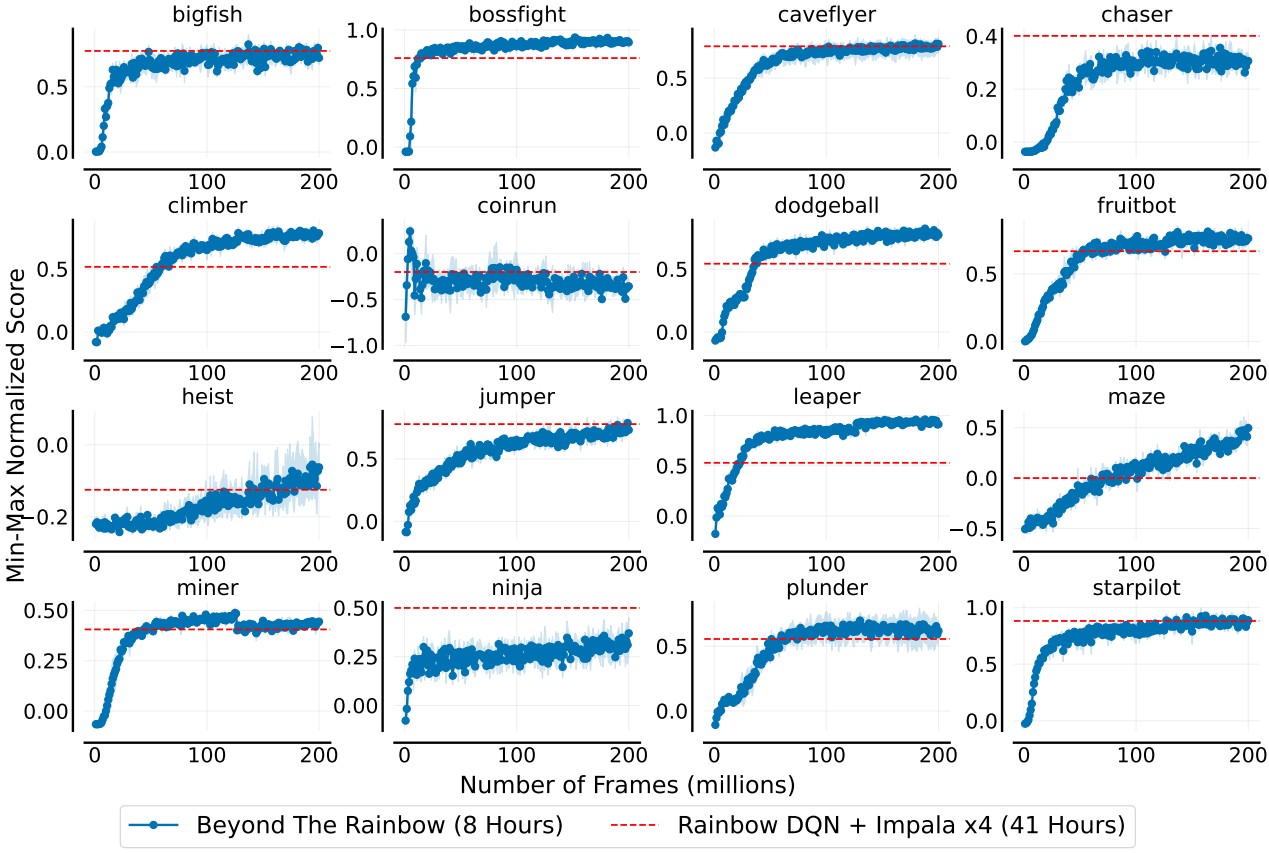

Figure B5: Performance of BTR on each individual game in the Procgen benchmark. Shaded areas show 95% confidence intervals. The red dotted line shows the performance of Rainbow DQN + Impala with 4x scaled Impala blocks (Cobbe et al., 2020), after 200M frames.

## C. Additional Ablations

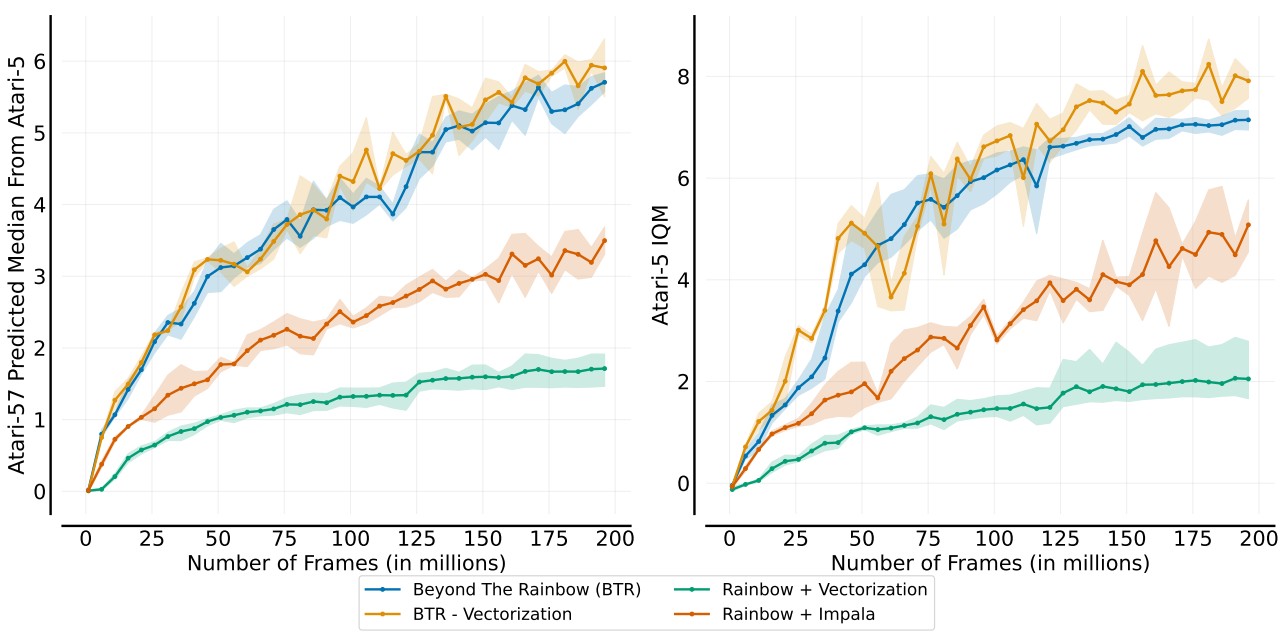

Figure C6: Figures show BTR's human-normalized scores without different components, with shaded areas showing 95% bootstrapped confidence intervals averaged over 4 seeds. **Left:** Predicted Atari-60 median score using the regression procedure defined in Aitchison et al. (2023). However, we find the predicted median does not match the true median (see Appendix K). **Right:** Interquartile mean across the 5 games.

## D. Hyperparameters

### D.1. Environment Details

Table D4: Environment Details for Atari Experiments.

| Hyperparameter | Value |
| --- | --- |
| Grey-Scaling | True |
| Observation down-sampling | 84x84 |
| Frames Stacked | 4 |
| Reward Clipping | [-1, 1] |
| Terminal on loss of life | False |
| Life Information | False |
| Max frames per episode | 108K |
| Sticky Actions | True |

Table D5: Environment Details for Procgen Experiments.

| Hyperparameter | Value |
| --- | --- |
| Grey-Scaling | False |
| Observation Size | 64x64 |
| Frames Stacked | 1 |
| Reward Clipping | False |
| Max frames per episode | 108K |
| Distribution Mode | Hard |
| Number of Unique Levels (Train & Test) | Unlimited |

## D.2. Algorithm Hyperparameters

Table D6: Table showing the hyperparameters used in the BTR algorithm.

| Hyperparameter | Value |
| --- | --- |
| Learning Rate | 1e-4 |
| Discount Rate | 0.997 |
| N-Step | 3 |
| IQN Taus | 8 |
| IQN Number Cos' | 64 |
| Huber Loss $\kappa$ | 1.0 |
| Gradient Clipping Max Norm | 10 |
| Parallel Environments | 64 |
| Gradient Step Every | 64 Environment Steps (1 Vectorized Environment Step) |
| Replace Target Network Frequency (C) | 500 Gradient Steps (32K Environment Steps) |
| Batch Size | 256 |
| Total Replay Ratio | $\frac{1}{64}$ |
| Impala Width Scale | 2 |
| Spectral Normalization | All Convolutional Residual Layers |
| Adaptive Maxpooling Size | 6x6 |
| Linear Size (Per Dueling Layer) | 512 |
| Noisy Networks $\sigma$ | 0.5 |
| Activation Function | ReLu |
| $\epsilon$-greedy Start | 1.0 |
| $\epsilon$-greedy Decay | 8M Frames |
| $\epsilon$-greedy End | 0.01 |
| $\epsilon$-greedy Disabled | 100M Frames |
| Replay Buffer Size | 1,048,576 Transitions ($2^{20}$) |
| Minimum Replay Size for Sampling | 200K Transitions |
| PER Alpha | 0.2 |
| Optimizer | Adam |
| Adam Epsilon Parameter | 1.95e-5 (equal to $\frac{0.005}{batchsize}$) |
| Adam $\beta1$ | 0.9 |
| Adam $\beta2$ | 0.999 |
| Munchausen Temperature $\tau$ | 0.03 |
| Munchausen Scaling Term $\alpha$ | 0.9 |
| Munchausen Clipping Value ($l_0$) | -1.0 |
| Evaluation Epsilon | 0.01 until 125M frames, then 0 |
| Evaluation Episodes | 100 |
| Evaluation Every | 1M Environment Frames (250K Environment Steps) |

## D.3. Clarity of the terms Frames, Steps and Transitions

Throughout the Arcade Learning Environment's history (ALE) (Bellemare et al., 2013; Machado et al., 2018), there have been many ambiguities around the terms: frames, steps and transitions, which are sometimes used interchangeably. Frames refer to the number of individual frames the agent plays, including those within repeated actions (also called frame skipping). This is notably different from the number of steps the agent takes, which does not include these skipped frames. When using the standard Atari wrapper, training for 200M frames is equivalent to training for 50M steps. Lastly, transitions refer to the standard tuple $(s_t, a_t, r_t, s_{t+1})$, where the timestep $t$ refers to a steps, not frames. We encourage researchers to make this clear when publishing work, including when mentioning the values of different hyperparameters.

# E. Beyond The Rainbow Architecture & Loss Function

## E.1. Architecture

Figure E7 shows the neural network architecture of the BTR algorithm. The architecture is highly similar to the Impala architecture (Espeholt et al., 2018), with notable exceptions:

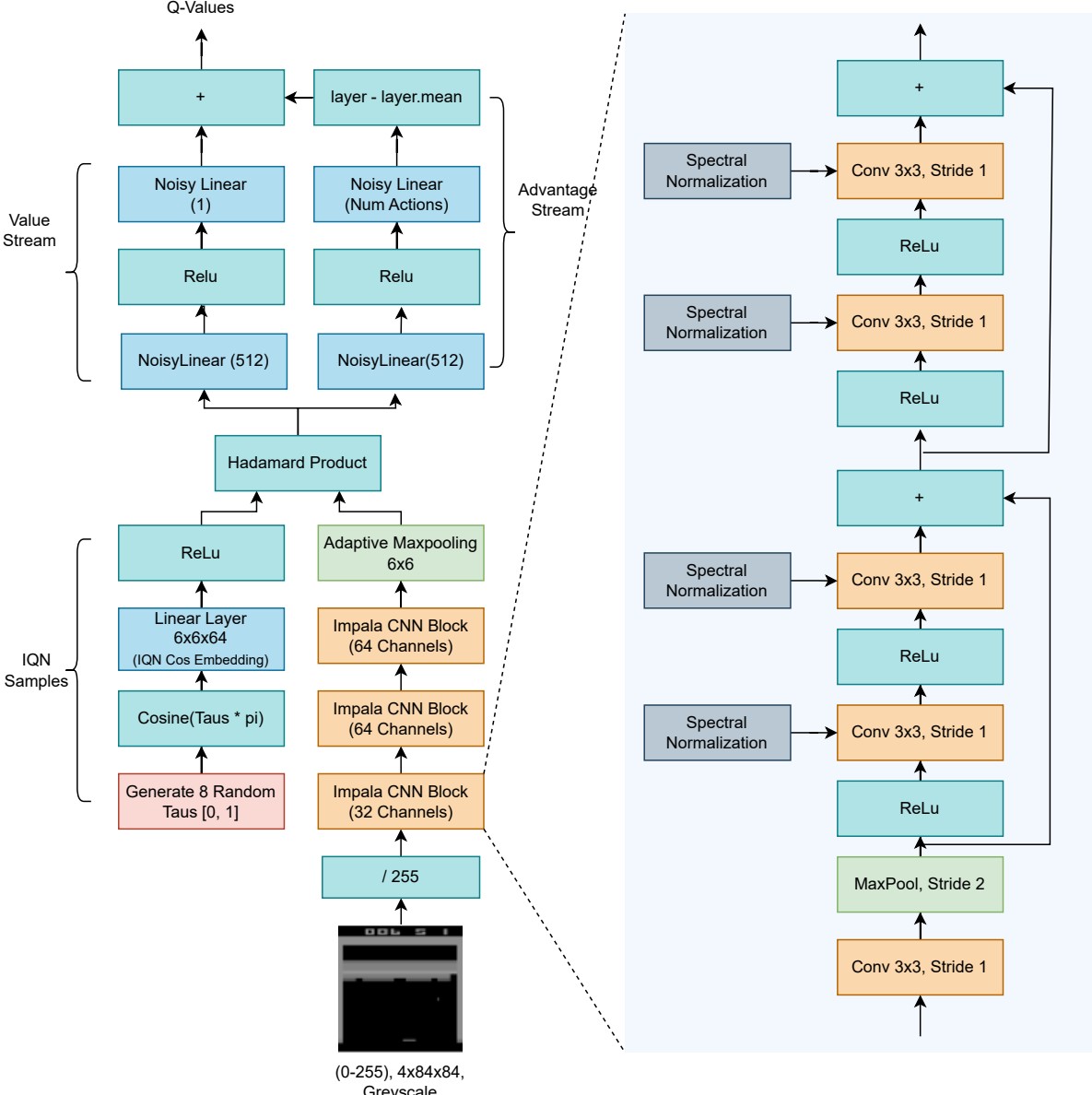

Figure E7: Architectural diagram of the BTR algorithm's neural network. The model contains a total of 2.91 million parameters, 2.52 million of which are within linear layers.

- **Spectral Normalization** Within each Impala CNN block, each residual layer (containing two Conv 3x3 + ReLu) has spectral normalization applied, as discussed in Section 3.1.

- **Maxpooling** Following the CNN blocks, a 6x6 adaptive maxpooling layer is added.

- **IQN** In order to use IQN, it is required to draw Tau samples, which are multiplied by the output of the CNN layers, as shown by the section 'IQN Samples' in figure E7.

- **Dueling** Dueling (as included in the original Rainbow DQN) splits the fully connected layers into value and advantage streams, where the advantage stream output has a mean of 0, and is then added to the value stream.

- **Noisy Networks** As included in Rainbow DQN, Noisy Networks replace the linear layers with noisy layers.

Lastly, the sizes of many of the layers given in Figure E7 are dependent upon the Impala width scale, of which we use the value 2. For example, the Impala CNN blocks have [16×width, 32×width, 32×width] channels respectively. The output size of the convolutional layers (including the maxpooling layer) is 6×6×32×width, as a 6x6 maxpooling layer is used. Lastly, the cos embedding layer, after generating IQN samples, requires the same size as the output of the convolutional layers. Hence, the size is selected accordingly. Another benefit of the 6x6 maxpooling layer is following the product of the convolutional layers and IQN samples, the number of parameters is fixed, regardless of the input size. Figure E8 shows the number of parameters the ablated versions of BTR have.

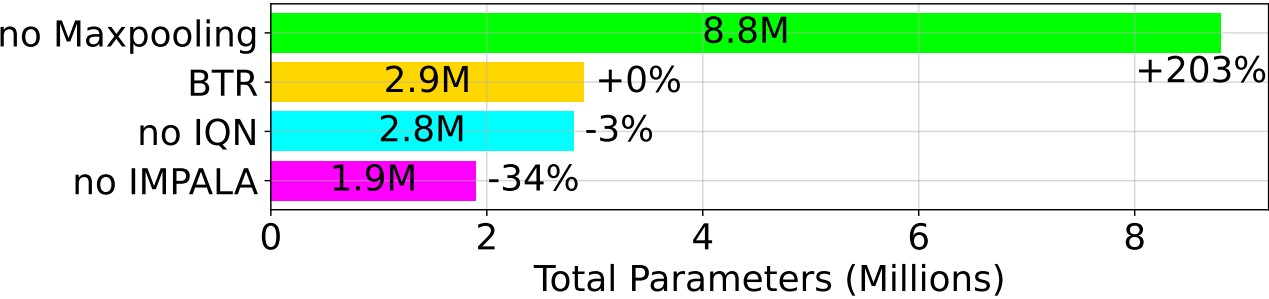

Figure E8: Total number of parameters in BTR with different components removed. Those not included in the graph (Munchausen and Spectral Normalization) used the same number of parameters as BTR.

### E.2. Loss Function

The resulting loss function for the BTR algorithm remains the same as that defined in the appendix of the Munchausen paper, which gave a loss function for Munchausen-IQN. As the other components in BTR do not affect the loss, the resulting temporal-difference loss function is the same. For self-containment, we include this loss function below:

$$TD_{BTR} = r_t + \alpha[\tau \ln \pi(a_t|s_t)]_{l_0}^0 + \gamma \sum_{a \in A} \pi(a|s_{t+1})(z_{\sigma'}(s_{t+1}, a) - \tau \ln \pi(a|s_{t+1})) - z_\sigma(s_t, a_t) \tag{E1}$$

with $\pi(\cdot|s) = sm(\frac{\tilde{q}(s,\cdot)}{\tau})$ (that is, the policy is softmax with q̃, the quantity with respect to which the original policy of IQN is greedy). It is also worth noting here that due to the character conflict of both Munchausen and IQN using $\tau$ (Munchausen as a temperature parameter, and IQN for drawing samples), we replace IQN's $\tau$ with $\sigma$. $l_0, \tau$ and $\alpha$ are hyperparameters set by Munchausen. We use the same values in BTR, also shown in our hyperparameter table in Appendix D.2.

### E.3. Analysis Confidence Intervals

Due to space constraints, we was unable to include confidence intervals for Table 2 in the main paper. A repeat of the main paper Table can be found in Table E7, with the associated confidence intervals in Table E8.

## F. BTR with and without Epsilon Greedy

One of the first observations we made early in the testing process was that the inclusion of using $\epsilon$-greedy in addition to NoisyNetworks benefited some environments but not others. Specifically, performance was reduced on *BattleZone* and *Phoenix*, both games where the agent reached very high levels of performance with extremely precise control. However, *DoubleDunk* performed significantly worse, only reaching a score of 0, rather than the score of 23 the final BTR algorithm achieved. Similar findings were also found in the full version of Rainbow DQN, which used only NoisyNetworks, which

Table E7: Repeat of the main paper's Table 2 for reference against the table of 95% confidence intervals below. Comparison of policy churn, action gaps, actions swaps and evaluation performance with different quantities of $\epsilon$-actions and color jitter (both only applied for evaluation). All measurements use the final agent, trained on 200 million frames, for Atari *Phoenix*, averaged over 3 seeds. Action Gap is the average absolute Q-value difference between the highest two valued actions. % Actions Swap is the percentage of times the agent's argmax action has changed from the last timestep. Policy churn is the percentage of states which the agent's argmax action has changed on after a single gradient step. Color jitter applies a random 10% change to the brightness, saturation and hue of each frame. For associated error with these values, please see Appendix E.3.

| Category | BTR | w/o Munchausen | w/o IQN | w/o SN | w/o Impala | w/o Maxpool |
|---|---|---|---|---|---|---|
| Action Gap | 0.282 | 0.055 | 0.180 | 0.274 | 0.215 | 0.264 |
| % Action Swaps | 36.6% | 47.7% | 42.2% | 40.3% | 41.1% | 39.3% |
| Policy Churn | 3.8% | 11.0% | 0.5% | 3.3% | 4.5% | 4.2% |
| Score ColorJitter | **212k** | 85k | 110k | 187k | 19k | 187k |
| Score $\epsilon = 0.03$ | **94k** | 42k | 62k | 75k | 10k | 86k |
| Score $\epsilon = 0.01$ | **194k** | 70k | 110k | 132k | 13k | 171k |
| Score $\epsilon = 0$ | 330k | 184k | 187k | 296k | 21k | **406k** |

Table E8: 95% confidence intervals for the main paper Table 2, calculated using 6 seeds. A repeat of the original table is shown above in Table E7.

| Category | BTR | w/o Munchausen | w/o IQN | w/o SN | w/o Impala | w/o Maxpool |
|---|---|---|---|---|---|---|
| Action Gap | [0.25, 0.31] | [0.05, 0.06] | [0.17, 0.19] | [0.25, 0.3] | [0.1, 0.33] | [0.23, 0.3] |
| % Action Swaps | [33.6, 39.6] | [45.8, 49.7] | [41.0, 43.3] | [38.5, 42.2] | [31.4, 50.8] | [38.2, 40.4] |
| Policy Churn | [3.3, 4.2] | [10.2, 11.7] | [0.5, 0.5] | [2.9, 3.6] | [3.9, 5.0] | [3.7, 4.7] |
| Score ColorJitter | [204k, 218k] | [77k, 92k] | [90k, 128k] | [173k, 201k] | [0k, 40k] | [158k, 215k] |
| Score $\epsilon = 0.03$ | [88k, 99k] | [38k, 46k] | [52k, 72k] | [68k, 82k] | [2k, 18k] | [74k, 97k] |
| Score $\epsilon = 0.01$ | [177k, 211k] | [62k, 77k] | [97k, 122k] | [139k, 176k] | [0k, 52k] | [143k, 199k] |
| Score $\epsilon = 0$ | [282k, 377k] | [116k, 251k] | [163k, 209k] | [244k, 348k] | [0k, 43k] | [332k, 479k] |

achieved a best score of -0.3 (Dopamine's "compact" Rainbow DQN, however, which did not use NoisyNetworks achieved 22). From this, we conclude that NoisyNetworks alone failed to sufficiently explore the environment, whereas $\epsilon$-greedy did not. From these results, we eventually decided to use both methods, but disable $\epsilon$-greedy halfway through training to reap the best of both techniques.

# G. Experiment Compute Resources

## G.1. Our Compute Resources

For running our experiments, we used a mixture of desktop computers and internal clusters. The desktop PCs used an GPU Nvidia RTX4090, CPU intel i9-14900k and 64GB of DDR5 6000mhz RAM. When using internal clusters, we used a mixture of GPUs, including Nvidia A100s, Nvidia Volta V100 and Nvidia Quadro RTX 8000. As for CPUs, we used 2 x 2.4 GHz Intel(R) Xeon(R) Gold 6336Y, 48 Cores. Lastly, we saved the models used to produce our analysis, totalling around 300gb across all of our ablations on the Atari-5 benchmark, saving a model every 1 million frames.

As most of our experiments were performed on desktop PC, in the main body of our paper we reference these speeds. We found that desktop PCs actually outperformed internal clusters, likely due to desktop CPUs being more suited to performing environment steps, outlined in the next subsection.

When testing ideas originally (those mentioned in Appendix H), we only tested them using a single run of the games *BattleZone*, *NameThisGame* and *Phoenix* unless otherwise stated. Whilst this method of evaluation is not statistically significant, for preliminary purposes with computational restrictions, we deemed this the best option.

### G.2. BTR with Different Hardware

In this work, we look to make high-performance RL more accessible to those with fewer computing resources, especially those only with access to desktop computers. Most of our experiments were performed with an RTX4090, we also provide some walltimes for 200M Atari frames for lower-end machines and provide a brief comparison of desktop PCs against internal clusters:

**Desktops:**

Original: RTX 4090, Intel i9-13900k (2023), 64GB RAM - **11.5 Hours**

RTX 3070, Ryzen 9 3900X (2019), 64GB RAM - **52 Hours**

RTX 2080 ti, Intel(R) Xeon(R) Silver 4112 CPU @ 2.60GHz (2018), 128GB RAM - **32 Hours**

**Internal Clusters:**

Nvidia H100, 48 Core Intel(R) Xeon(R) Platinum 8468 (2023), 2TB RAM - **15 Hours**

Nvidia A100, 24 Core Intel(R) Xeon(R) Gold 6336Y (2021), 512GB RAM - **22 Hours**

We note that there is significant variability in hardware (processors, memory bus speeds, etc), but the results still show reasonable times compared to not using BTR. Overall, we found that training BTR was very capable of running on lower end machines, with the agent (excluding the environments) using around 15GB of RAM. The main performance bottleneck was running the environment in parallel, making the number of CPU cores and processor speed most important. BTR also provides strong performance long before 200M frames, thus providing practical utility for lower-end machines.

## H. Other Things We Tried

Throughout the development of the BTR algorithm, we experimented with many different components and hyperparameters. A brief list of ideas we tried that performed worse or equivalent to the final algorithm includes:

- Using Exponential Moving Average networks rather than using fixed target networks (this was both computationally slower and performed worse).

- Varying the frequency of updating the target network (we tested 250, 500 and 1000, finding 500 to perform best).

- Changing the size of maxpool layer following the convolutional layers (we tested 4 and 8, however 6 performed significantly better).

- Decaying the learning rate from $1 \times 10^{-4}$ to 0 over the course of training (this made no significant difference).

- Different learning rates, finding $1 \times 10^{-4}$ to perform best, however $5 \times 10^{-5}$ also performed similarly as was used in Implicit Quantile Networks (IQN).

- Using the AdamW optimizer(Loshchilov & Hutter, 2019) which uses weight decay with the decay parameter $1e - 4$, however found this made no significant difference.

- Using the GeLu activation instead of ReLu, which drastically reduced performance.

Only testing on a single environment (*BattleZone*), we also tried:

- Annealing the discount rate from 0.97 to 0.997 throughout training, but found no significant difference.

- Applying spectral normalization to the linear layers (dramatically worse performance).

- Increasing the number of cos' from IQN (no significant difference on performance).

- Using Dopamine's Prioritized Experience Replay buffer which doesn't include a $\alpha$ value (moderately worse performance).

- As discussed in F, we also tried not using $\epsilon$-greedy when using noisy nets.

Lastly we also tried removing some of the original components from Rainbow DQN on Atari *BattleZone*, including Dueling, Prioritized Experience Replay and Noisy Networks. Prioritized Experience Replay and Noisy Networks both proved beneficial, so were kept in the algorithm. Dueling did not seem to make any significant difference, however we did not choose to remove it for a clearer continuation of Rainbow DQN, in addition to potentially being useful in other Atari environments.

Shortly after the submission of this work, we tested BTR with addition of Layer Normalization, and found positive results. Layer Normalization can improve the robustness to a variety of pathologies that cause loss of plasticity (Lyle et al., 2024), and helps to improve the conditioning of the network's gradients in RL (Ball et al., 2023). Below in Figure H9 we show the impact of including layer normalization into BTR.

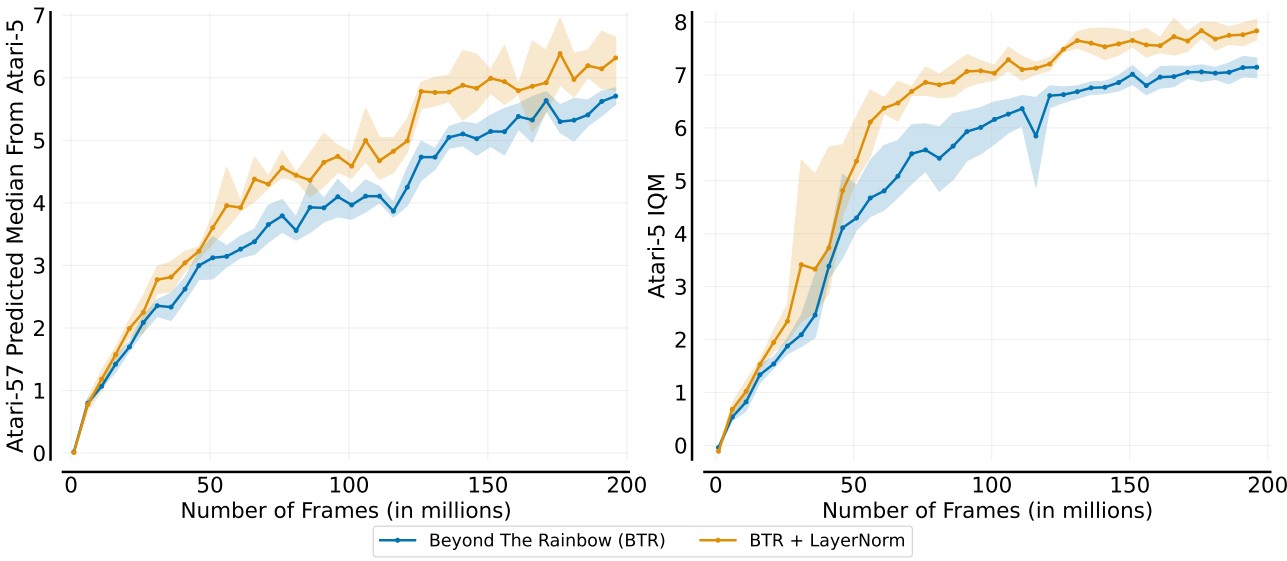

Figure H9: Graph shows Atari-5 performance with and without layer normalization using Inter-quartile mean and Atari-60 predicted median from Aitchison et al. (2023). Layer normalization uses 3 seeds, with shaded areas showing 95% confidence intervals.

Table H9: Maximum scores obtained during training (averaged over 100 episodes and all performed random seeds) after 200M Frames on the Atari-5 Environment, compared to BTR with Layer Normalization.

| Game | Random | Human | BTR + Layer Normalization | BTR |
|---|---|---|---|---|
| BattleZone | 2360 | 37188 | **183240** | 168340 |
| DoubleDunk | -19 | -16 | **23** | **23** |
| NameThisGame | 2292 | 8049 | **33258** | 27917 |
| Phoenix | 761 | 7243 | **493762** | 427481 |
| QBert | 164 | 13455 | **47384** | 42927 |
| IQM | 0.000 | 1.000 | **8.191** | 7.739 |
| Median | 0.000 | 1.000 | **5.379** | 4.766 |
| Mean | 0.000 | 1.000 | **20.836** | 18.453 |

# I. Altered Atari Environment Settings

In order to investigate the impact of the environmental sticky actions parameter and to compare against other works, we include results for it on the Atari-5 benchmark in Figure I10.

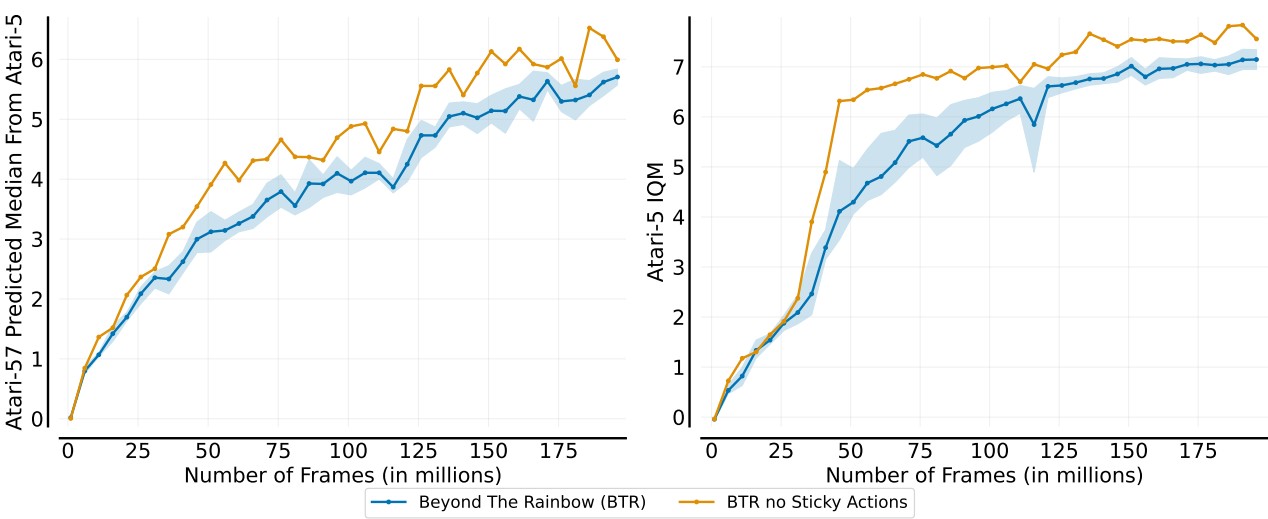

Figure I10: Graph shows Atari-5 performance with and without sticky actions (sticky actions is the default) using Inter-quartile mean and Atari-60 predicted median from Aitchison et al. (2023). No Sticky Actions uses a single seed, so this result should be used with caution.

Some prior works choose to pass life information to the agent (Schmidt & Schmied, 2021). To clarify, this is different to terminal on loss of life. Life information does not reset the episode upon losing a life, but does pass a terminal to the buffer, allowing the agent to experience further into episodes while also giving the agent a negative signal for losing a life. This setting is not recommended in Machado et al. (2018), and works which use it are **not** comparable to those which don't. To emphasize this point, we take the three games from the Atari-5 and perform a comparison.

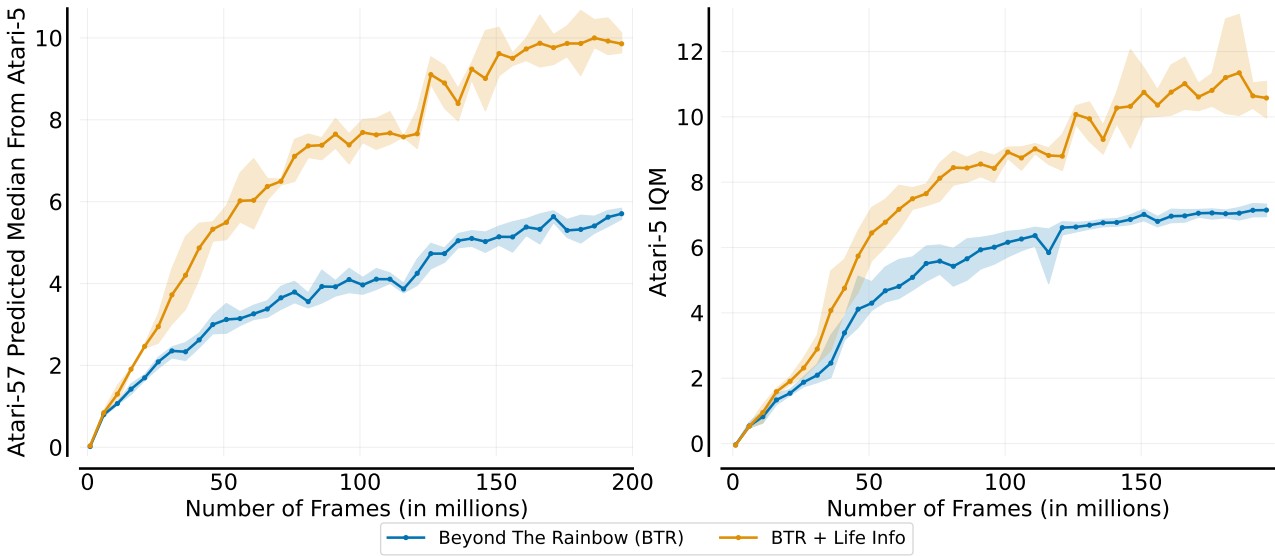

Figure I11: Graph shows Atari-5 performance with and without life information using Inter-quartile mean and Atari-60 predicted median from Aitchison et al. (2023). Life Information uses 3 seeds, with shaded areas showing 95% confidence intervals. From this we conclude results using life information are invalid for comparison.

## J. BTR for Wii Games

BTR interfaces with different Wii Games via the Dolphin Emulator. Specifically, we use a forked repository of Dolphin Emulator to allow Python scripts to interact with the emulator. This includes loading savestates (used to reset episodes),

grabbing the screen as a PIL image at the Wii's internal resolution of 480p (downsampled to 140x114 and grey-scaled, used for all observations), reading the Wii's RAM (used for reward functions and termination conditions) and allowed programmatic input into the emulator (for setting actions). Using Dolphin's portable setting, we are able to run multiple Dolphin Emulators simultaneously on the same machine. Each instance runs as a unique process, and communicates with the agent via Python's multiprocessing library. Similarly to the Atari benchmark, for all games we used a frameskip of 4.

### J.1. Super Mario Galaxy

This environment used Super Mario Galaxy's final level, *The Center of the Universe*, and had to make it to the final fight at the end of the game. The agent had 6 actions, including None, moving in each direction and jumping. Additionally, if the jump action was performed following a movement, the agent would continue to move in that direction.

Rewards were given via finding many values in the Wii's memory that resembled progress in the level. The agent was then rewarded for this progress value increasing from the last frame. If the agent's position entered a set region, the progress variable would be moved. Additionally, the game uses a life system, where the player has a maximum of 3 lives and can lose or gain lives in many different ways. The agent was given a reward of +1 for gaining a life, and -1 for losing a life. Lastly, episode termination occurred if the agent reached 0 lives, or if the agent made it to the end of the level. For this task, we also allowed the agent to start episodes at many points throughout the level, which rapidly sped up training since the agent could easily experience different areas of the level.

Whilst a difficult task, once the agent first completed the level, it did not take long to start consistently completing it due to the deterministic nature of the game.

### J.2. Mario Kart Wii

The Mario Kart Wii environment had the agent play against the game's internal opponents (on hard mode, with 12 racers including the agent), on the course Rainbow Road (with items on the 150cc speed setting). The agent had to complete 4 laps of the course to finish the race. The agent had just 4 action, including accelerate, drifting left or right, and using its item. While this limited the agent's potential actions substantially, we found using fewer actions to dramatically accelerate training.

Rewards of +1 were given via reaching checkpoints that were scattered throughout the course (100 in total per lap). Additionally, if the agent's speed dropped below a set threshold (65 km/h), the agent would receive a reward of -0.01 per frame. The agent would be terminated with a reward of -10 if its speed dropped below the threshold for over 80 frames, or with a reward of +10 for finishing the race, with a bonus based on the position the agent finished in. Lastly, the agent was rewarded with a +1 for using its item. Without this reward, we found the agent to often neglect using its item, likely due to many of the items only providing rewards in the long term, such as slowing down other racers or blocking incoming items far in the future. Similarly to Super Mario Galaxy, we had the agent start the episode in multiple positions around the first lap, allowing it to experience the whole track early in training.

This agent took the longest to train, taking around 160M frames to reach consistent completion. In particular, the agent took a long time to consistently complete the race due to the other racers and randomized items making the environment highly stochastic, with many rare scenarios which could cause the episode to terminate.

### J.3. Mortal Kombat

The Mortal Kombat environment put the agent in the game's *endurance* mode, where the agent would sequentially fight 15 different opponents, but keep retain its health between fights, and only gain health after defeating every 3 opponents. We provided the agent with 14 actions, including: None, Left, Right, Up, Down, Axe Kick, Punch, Snap Kick, Grab, Block, Toggle Weapon, Jump Left, Jump Right, and Crouch. These actions were far from the game's total action space, and limited the agent's ability to perform some of the combos within the game. We limited the agent's actions as the full action space is extremely large.

The agent was positively rewarded for damaging the opponent, and negatively rewarded for taking damage, with one taking one tenth of the health bar equating to +1 reward respectively. The episode was terminated with a reward of -10 for reaching 0 health, and +10 for defeating the 15th and final enemy.

The Mortal Kombat agent learned considerably faster than Super Mario Galaxy and Mario Kart Wii, first completing the

environment in 50M frames, and getting progressively more consistent until training was stopped at 90M frames. The agent quickly learned how to dodge enemy hits, and relied heavily upon this strategy.

## K. Atari-5 Regression Procedure

In our main paper ablation figure (Figure 5), we used the regression procedure recommended in Atari-5 (Aitchison et al., 2023). This procedure is typically used to predict the Median score across the entire 57 game Atari suite, while only needing to use 5 games. While we believe this procedure produces a valid and useful plot, we find that BTR did differ significantly from the predicted value. We opted to use both the predicted median and the IQM across the 5 games to give two easy to interpret averages. Figure K12 shows the 57 game suite's true median, compared to the median predicted by Atari-5.

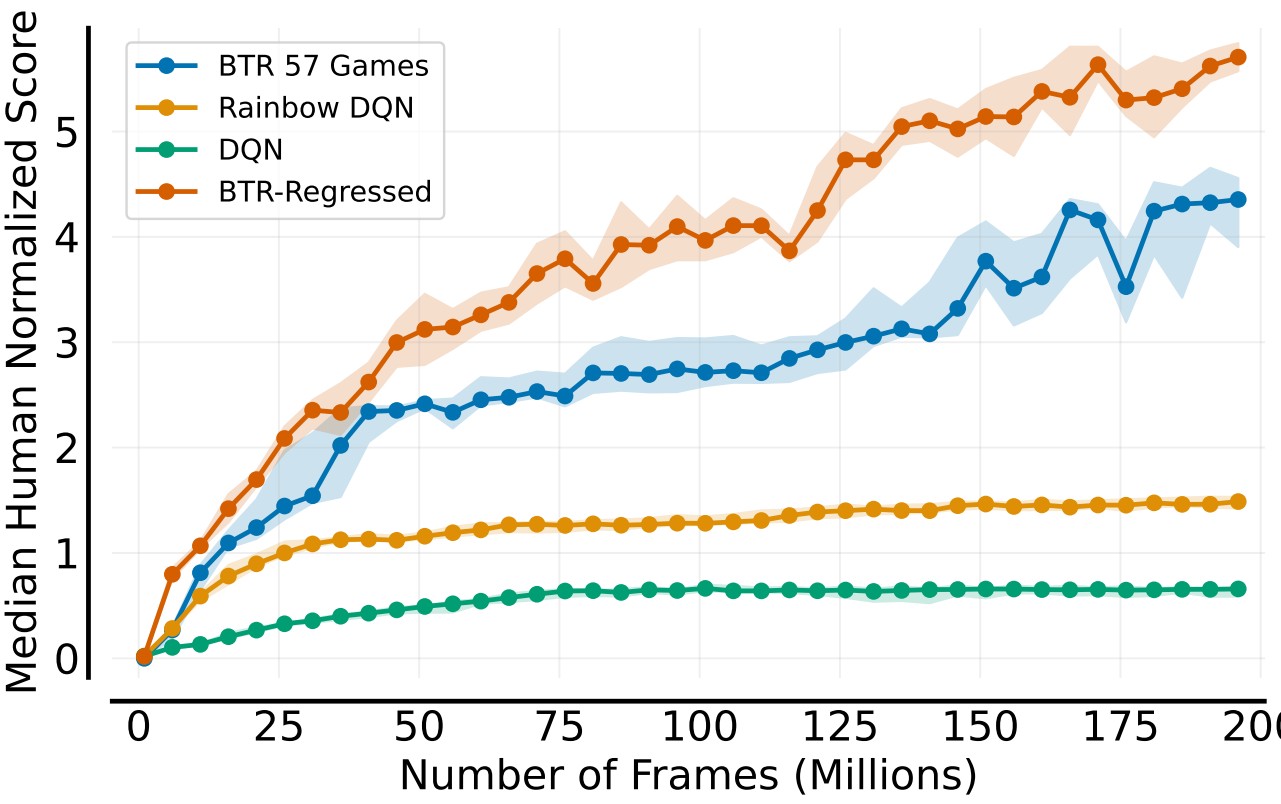

Figure K12: BTR's 60 game median against that predicted by Atari-5 using the regression procedure from Aitchison et al. (2023). Shaded areas show 95% confidence intervals using 4 seeds.

