# OpenReview forum: "Beyond The Rainbow: High Performance Deep Reinforcement Learning on a Desktop PC"
_ICML.cc/2025/Conference — ICML 2025 poster_

### Official Review · Reviewer_SiM3 · 2025-03-03

**Overall Recommendation:** 3

**Summary:**

Similar to Rainbow DQN, this paper integrates several improvements from existing RL literature to improve the performance of reinforcement learning agents. The authors show that their ensemble method BTR achieves state-of-the-art performance on Atari tasks using 200M frames on a desktop PC. The authors also show performance on other complex 3D games such as Super Mario Galaxy, Mario Kart, and Mortal Kombat. The ablation experiments in the paper show the impact of each improvement proposed.


## Update after rebuttal
During the rebuttal session, since the authors have added more recent baseline results that greatly improve the persuasiveness of the paper, I now tend to weak accept this paper. Good work!

**Claims And Evidence:**

As the title and experiments show, this paper mainly proposes an integrated method that integrates multiple existing reinforcement learning technologies, and proves through experiments that it has achieved SOTA performance on 200M Atari (mainly) and can run on personal PCs.

However, the methods mainly compared in the paper, such as Rainbow, DQN, etc., may be far from the current SOTA algorithms. Even if BTR can surpass them, it is difficult to prove its SOTA performance, and thus it is difficult to prove the main contribution of this paper (i.e. the advantages of integrating multiple RL technologies). If the author can give a comparison with more recent papers, it may greatly enhance the persuasiveness of the paper, such as MEME [1], EfficientZero [2], etc.

[1] Human-level Atari 200x faster

[2] Mastering Atari Games with Limited Data

**Essential References Not Discussed:**

Since the main contribution of the paper is to propose a set of practical and efficient RL algorithms, this paper lacks comparison with some recent existing efficient RL algorithms, which at least include MEME, EfficientZero, etc.

**Experimental Designs Or Analyses:**

The experiment in this paper mainly focuses on using RL algorithms to train multiple video game agents, and mainly compares evaluation metrics  such as IQM, Mean HNS, and Median HNS. However, the main problem may be the lack of more recent and more convincing baselines.

In addition, the method in the paper seems to use life info from Atari, so what is the performance without using life info?

**Methods And Evaluation Criteria:**

The proposed method is an integration of several existing improvements (at the algorithm level). The main experiments were conducted on the 200M Atari task. The evaluation metrics used in the experiment, such as IQM, are reasonable. However, it is worth noting that the baseline method compared in this paper may be far from representing the current SOTA method on the 200M Atari task. As mentioned in (Claims And Evidence), there are at least more recent baselines like MEME and EfficientZero. If the author can add corresponding comparisons, it may be more convincing.

Furthermore, the ability to run RL algorithms on personal PCs is certainly exciting, but its practical implications may still require extensive discussion and clarification.

**Other Comments Or Suggestions:**

If the author can compare it with some more recent RL algorithms, it will greatly enhance the persuasiveness of the paper.

**Other Strengths And Weaknesses:**

The main advantage of this paper is that it proposes a set of efficient RL algorithms that can run on high-performance personal PCs, but this paper lacks comparison with the key latest related literature, and also lacks more convincing applications of BTR method in PC.

**Questions For Authors:**

1. How does BTR compare with newer RL algorithms on the 200M Atari task? For example, MEME, EfficientZero, etc.
2. In addition to video games, can the author provide other more convincing application scenarios for BTR?
3. On the Atari task, how does BTR perform without using life info?
4. In addition to the integration of multiple methods, can the author prove its original contribution?

**Relation To Broader Scientific Literature:**

Since the paper mainly conducts experiments on Atari tasks, it mainly compares a series of other existing RL algorithms, such as Rainbow, Impala, DQN, etc. However, there is still a lack of more recent paper comparisons, such as MEME, EfficientZero, etc.

**Theoretical Claims:**

This paper is mainly an experimental paper, that is, to verify the effectiveness of the proposed algorithm, and there are no outstanding theoretical results in the main body of the paper.

---

> ### Author Rebuttal · Authors · 2025-03-28
>
> We thank the reviewer for their concise review.
>
> **Life Information** - We do not use life information in any main paper results, as we specify in Section 4.1. Furthermore,  we provide a detailed comparison of BTR with and without life information in Appendix I. If your low score was due to doubts about our empirical performance stemming from this matter, we hope you are willing to reconsider now that this has been clarified.
>
> **Comparison against baselines** - We already compare against SoTa algorithms for Atari 200M in Table 3. We did not include such baselines in Figures 1 and 2 as we wanted to limit this comparison to those algorithms which can be run with widely accessible compute power. However, we are willing to replace Figure 1 with a walltime efficiency curve against Dreamer v3, as we feel this better conveys the message of the paper (MEME did not release this data). We can also add Dreamer v3 and MEME to Figure 2, with a note indicating their walltime to allow for a fairer comparison as we believe that with the appropriate context this will improve the paper. We do, however, argue against the inclusion of sample efficient algorithms (Atari-100K), as these algorithms use drastically different resources (100K frames vs 200M), and critically are benchmarked on a 26-game subset, rather than the full suite, making for an unfair comparison. If you believe this to be pivotal, we can include a figure in the appendix for the 26-game subset. Below is a comparison against SoTa algorithms on the mentioned 26-game subset:
>
> | Algorithm                             | Frames | A100 Walltime        | IQM (26-Game Subset) |
> |--------------------------------------|---------------|----------------|----------------------|
> | MEME                                | 200M | Not Reported*            | 18.491               |
> | Dreamer v3                           | 200M | 7.7 Days                | 14.305               |
> | Beyond The Rainbow (BTR)            | 200M | 22 Hours                | 11.202               |
> | PQN [1]                                  | 400M | 1 Hour                | 5.014**                |
> | EfficientZero v2                     | 100K | 2.7 Hours***            | 1.305                |
> | BBF                                  | 100K | 7.8 Hours               | 1.045                |
> | Dreamer v3                           | 100K | 2.4 Hours               | 0.543                |
>
>
> *MEME used a shared server with a TPUv4.
>
> **PQN used life information, making its results appear significantly higher.
>
> ***EfficientZero was tested on a server with 8 RTX 3090s, not an A100.
>
>
> **Practical Implications of RL on Desktop PCs** - Could you further clarify this point? We are unsure exactly what you are asking.
>
> [1] Gallici, Matteo, et al. "Simplifying deep temporal difference learning." arXiv preprint arXiv:2407.04811 (2024).

---

> > ### Comment · Reviewer_SiM3 · 2025-04-04
> >
> > Thank you for the author's detailed response, which addresses my concerns. Adding  more recent baseline results can indeed greatly improve the persuasiveness of the paper, so I am happy to update my evaluation to weak accept.

---

> > > ### Author Response · Authors · 2025-04-07
> > >
> > > We would like to thank the reviewer for the insightful discussion and raising of their score.
> > >
> > > Additionally, we would like to clearly set out our key contributions and applications as per your request. Aside from integrating, testing, tuning and ablating a wide variety of different components, we believe the final algorithm is a novel contribution as it provides a practical algorithm for our targeted audience of academics, hobbyists and students. Beyond this, our analysis section provides new insights into many different components which were not explored in the respective papers. Furthermore, our appendices provide important information to the community for applying RL algorithms, such as the impact of life information (and why using it invalidates results), and how we designed MDPs for the Wii games.
> > >
> > > As for the practical implementations of BTR besides games, in its current state the algorithm can be used for any image-based, discrete action environment. This has numerous potential applications such as robotics, healthcare and UAVs. BTR also presents future work to be combined with other areas of RL such as value-based methods for complex action spaces [1, 2], Sim-to-real [3, 4] and offline-RL [5, 6] making for a more widely applicable algorithm and widely usable algorithm.
> > >
> > > [1] Tavakoli, Arash, Fabio Pardo, and Petar Kormushev. "Action branching architectures for deep reinforcement learning." Proceedings of the aaai conference on artificial intelligence. Vol. 32. No. 1. 2018.
> > >
> > > [2] Tavakoli, Arash, Sina Ghiassian, and Nemanja Rakićević. "Learning in complex action spaces without policy gradients." arXiv preprint arXiv:2410.06317 (2024).
> > >
> > > [3] Wagenmaker, Andrew, et al. "Overcoming the Sim-to-Real Gap: Leveraging Simulation to Learn to Explore for Real-World RL." Advances in Neural Information Processing Systems 37 (2024): 78715-78765.
> > >
> > > [4] Zhao, Wenshuai, Jorge Peña Queralta, and Tomi Westerlund. "Sim-to-real transfer in deep reinforcement learning for robotics: a survey." 2020 IEEE symposium series on computational intelligence (SSCI). IEEE, 2020.
> > >
> > > [5] Prudencio, Rafael Figueiredo, Marcos ROA Maximo, and Esther Luna Colombini. "A survey on offline reinforcement learning: Taxonomy, review, and open problems." IEEE Transactions on Neural Networks and Learning Systems (2023).
> > >
> > > [6] Ada, Suzan Ece, Erhan Oztop, and Emre Ugur. "Diffusion policies for out-of-distribution generalization in offline reinforcement learning." IEEE Robotics and Automation Letters 9.4 (2024): 3116-3123.

---

### Official Review · Reviewer_sM3r · 2025-03-12

**Overall Recommendation:** 3

**Summary:**

This paper proposes Beyond the Rainbow (BTR), an algorithmic successor to Rainbow DQN that improves asymptotic performance, data-efficiency, and wall-time efficiency through a series of algorithmic and architectural modifications informed by existing but recent literature. Key changes include using an Impala backbone rather than the traditional 3-layer Nature CNN, adaptive maxpooling, spectralnorm, IQN value estimation instead of C51, Munchausen RL for TD-learning instead of Double DQN, and finally use of vectorized environments and hyperparameter-tuning for improved wall-time efficiency. Experiments are conducted on Atari as well as a number of 3D video game environments, and the authors demonstrate that BTR performs significantly better than Rainbow DQN and a number of other model-free baselines.

## Post-rebuttal assessment

As mentioned in my rebuttal reply, I will maintain my score of weak accept but appreciate the authors' response to my comments as well as those of my fellow reviewers.

**Claims And Evidence:**

The main claim of the paper is that the algorithmic and architectural improvements upon Rainbow DQN lead to a new state-of-the-art for model-free (non-recurrent) RL on the chosen domains. The proposed changes are well motivated and validated through a series of ablation studies that provide empirical evidence for the claims. Experiments establish that BTR outperforms Rainbow DQN by a large margin on each of the chosen domains.  I appreciate the inclusion of a number of unconventional domains for evaluation (the 3D environments); they all look fairly challenging and represent real use cases for RL in the context of video games. My main issue with the experimental results is that all of the baselines are considered rather old and outdated. For example, the authors mainly compare against Rainbow DQN (2018), as well as methods for which numbers are reported in RLiable (2021). There has been substantial progress in the area of RL in the past 4 years, which is largely ignored in the experimental evaluations. This makes the claim that BTR achieves a new state-of-the-art somewhat shaky. Given that most RL algorithms for discrete action spaces report benchmark numbers for Atari (either 100k or 200M) I see little reason not to include more recent results, such as DreamerV3 (2023) as opposed to DreamerV2 (2020). If I read the results correctly, it appears that DreamerV3 achieves 8.30 human-normalized median score on Atari 60 vs. 4.69 for the proposed method, and 6.93 for MuZero [1]. While I undoubtedly find the proposed method appealing due to its wall-time efficiency, I feel that not positioning the results wrt any RL results more recent than 2021 is highly problematic. I include two model-based RL results here to prove my point, but I'm sure that there are more recent model-free results readily available for Atari 200M as well.

[1] Hafner et al., DreamerV3: https://arxiv.org/abs/2301.04104 (2023)

**Essential References Not Discussed:**

I am not very familiar with recent literature on model-free RL algorithms for Atari 200M, but there has been substantial work on model-based algorithms for Atari 200M (MuZero and DreamerV3 come to mind [1]), as well as both model-free (DrQ, SPR, BBF) [2] and model-based (EfficientZero, EfficientZero-V2) [3] algorithms for Atari 100k. The paper would benefit from discussion and empirical comparison to more recent literature.

[1] Hafner et al., DreamerV3: https://arxiv.org/abs/2301.04104 (2023)

[2] Schwarzer et al., BBF: https://arxiv.org/abs/2305.19452 (2023)

[3] Wang et al., EfficientZero-V2: https://arxiv.org/abs/2403.00564 (2024)

**Ethical Review Concerns:**

No notable concerns

**Experimental Designs Or Analyses:**

Experimental design appears solid. I have concerns regarding baselines, as previously discussed.

**Methods And Evaluation Criteria:**

Yes, I believe that the chosen domains are appropriate for benchmarking. I have concerns about the choice of baselines, as discussed above.

**Other Comments Or Suggestions:**

The paper has a few minor typos but they do not detract from my understanding in any meaningful way.

**Other Strengths And Weaknesses:**

The paper is generally well written, the proposed algorithmic and architectural changes are well motivated, and ablations provide insights into the relative importance of each design choice. I appreciate the use of RLiable metrics.

**Questions For Authors:**

I would like the authors to provide justification for their choice of baselines (in particular the wrt omission of more recent literature) as well as a general lack of discussion of newer literature. I have provided some references but hope that the authors can conduct a more thorough literature study as well given their focus on model-free methods for Atari 200M in particular.

It would be helpful to provide additional model-free baselines for the 3D video game domains but I understand that this may not be possible within the strict time frame of a rebuttal. I highly recommend comparison to a more recent algorithm for a future revision.

**Relation To Broader Scientific Literature:**

The contributions are well motivated and positioned wrt. previous work. There is a serious lack of discussion and comparison to more recent literature; the chosen baselines are 3-12 years old.

**Theoretical Claims:**

Not relevant. Paper is empirical in nature.

---

> ### Author Rebuttal · Authors · 2025-03-28
>
> Thank you for your detailed and constructive review, and for your appreciation of the evaluation on unique environments.
>
> **Comparison against baselines** - We already provided a discussion on SoTa algorithms for Atari 200M in Table 3, and two of the listed algorithms (BBF and EfficientZero V2) are sample sample efficient algorithms which only used 100K frames rather than 200M. Furthermore, they were never benchmarked on the full Atari set, but rather the reduced 26 game sample efficient benchmark. Therefore, we cannot include these in Figures 1 and 2. We did not include MEME and Dreamer v3 in Figures 1 and 2 as we did not want to present an ‘apples to oranges’ comparison, since these SoTa algorithms are inaccessible to our targeted users due to heavy compute requirements. However, to avoid any confusion around BTR’s performance, we are willing to replace Figure 1 with a walltime efficiency curve that includes Dreamer v3, as we feel this better conveys the message of the paper. We can also add Dreamer v3 and MEME to Figure 2, with a note indicating their walltime to allow for a fairer comparison as we believe that with the appropriate context this will improve the paper. If this is your main critique of the paper, given that this is simple to remedy, we hope it may be grounds for improving your score.
>
> | Algorithm                             | Frames | A100 Walltime        | IQM (26-Game Subset) |
> |--------------------------------------|---------------|----------------|----------------------|
> | MEME                                | 200M | Not Reported*            | 18.491               |
> | Dreamer v3                           | 200M | 7.7 Days                | 14.305               |
> | Beyond The Rainbow (BTR)            | 200M | 22 Hours                | 11.202               |
> | PQN [1]                                  | 400M | 1 Hour                | 5.014**                |
> | EfficientZero v2                     | 100K | 2.7 Hours***            | 1.305                |
> | BBF                                  | 100K | 7.8 Hours               | 1.045                |
> | Dreamer v3                           | 100K | 2.4 Hours               | 0.543                |
>
>
> *MEME used a shared server with a TPUv4.
>
> **PQN used life information, making its results appear significantly higher.
>
> ***EfficientZero was tested on a server with 8 RTX 3090s, not an A100.
>
> **More Baselines for Wii Games** - As you mention, unfortunately this may not be possible during the rebuttal period. Out of interest, what algorithm would you like to see? We found other SoTa algorithms such as Dreamer v3 to be too computationally expensive to run on more demanding environments.
>
> [1] Gallici, Matteo, et al. "Simplifying deep temporal difference learning." arXiv preprint arXiv:2407.04811 (2024).

---

### Official Review · Reviewer_eCYf · 2025-03-16

**Overall Recommendation:** 3

**Summary:**

Rainbow now for few years has been a SoTA DQN based RL method. In this paper, authors redo the basic idea behind the Rainbow and collect a new, since Rainbow, a set of tips and tricks and include them into Rainbow, thus obtaining BTR, Beyond-the-Rainbow. They mostly evaluate performance of the completed system with ALE, but also run some experiments with three games from WII and some Procgen experiments.

## update after rebuttal

I appreciate authors rebuttal, but taking into account rebuttal and other reviewers opinions I will keep my score.

**Claims And Evidence:**

Claims are convincingly supported by evidence.

**Essential References Not Discussed:**

-

**Experimental Designs Or Analyses:**

Yes I did check and no issues.

**Methods And Evaluation Criteria:**

Mostly makes sense, but the focus on wallclock time instead of environment steps can be misleading. Wallclock time measurement favors the use of vectorization, which is useful if the goal is to engineer a system that is able to get more bang out of the hardware, but I would argue that most RL researchers are interested in sample efficiency and there wallclock time is misleading as is the number of gradient steps. This point obviously does not discount the usefulness of vectorization that is amply demonstrated by this paper.

**Other Comments Or Suggestions:**

-

**Other Strengths And Weaknesses:**

It is clear that this paper is very useful for the RL researchers and practitioners. It contains an ample evidence on what tricks together are the most useful. I am sure the emplrical results in this paper will be greatly studies and scrutinized. Authors have taken their time to look at the results from many different points of view, I find Table 2 to be very instructive in terms of really evaluating the effectiveness of respective tricks.

This paper is styled from the benchmarking point of view, where idea seems to be to get the most out of ALE. This itself is not necessarily a bad thing as it is useful know what works and does not work with respect to experimenting with ALE. But it is known to have serious issues, such as non-stochasticity that is fixed in Procgen. But in this paper authors run only cursory experiments with Procgen, I would have liked to see maybe the roles reversed where Procgen is the focus and ALE is shown for benchmarking reasons. Finally, three games using Delphi, WII games, are just a curioisity. So, to strenghen the scientific message of this paper I would suggest to refocus this paper for Procgen.

Continuing from the previous point, sparse reward scenarious would benefit from better sample efficiency, so environment that emphasizes it would then be most useful in this respect. One option is to use some scenarious in the MiniWorld.

**Questions For Authors:**

- I am curious about the Fig 1, why there is a bump in the BTR curve?

**Relation To Broader Scientific Literature:**

-

**Theoretical Claims:**

No theoretical claims.

---

> ### Author Rebuttal · Authors · 2025-03-28
>
> We thank the reviewer for their review and their appreciation of how our work will be useful for both researchers and practitioners.
>
> **Steps vs Walltime** – We feel the reviewer has somewhat misunderstood the main point of this paper. As stated in the title, abstract, and contributions, the purpose of our work is to create a high-performance algorithm that is computationally accessible—something that is poorly measured by environment steps. For example, while BTR takes approximately 12 hours to process 200 million frames, Dreamer v3 takes almost 8 days to do the same and requires considerably more RAM and GPU memory, illustrating that environment steps are a poor metric for computational accessibility.
>
> **Use of the ALE** – The ALE remains the most widely used RL benchmark for good reason, and we feel it is being unfairly dismissed here. The standard evaluation protocol for the ALE [2] (which we follow) includes both no-op starts and sticky actions, which are explicitly intended to prevent agents from exploiting determinism in the environment. Moreover, the ALE gained its popularity due to its extremely diverse set of tasks, a strength we believe is being overlooked. Other SoTa algorithms, such as MEME [3], benchmark exclusively on Atari for this very reason. The ALE also contains numerous sparse-reward environments, which can be found in our Atari-60 results.
>
> **Bump in Figure 1** – As discussed in Appendix F, this bump is due to epsilon-greedy exploration being disabled partway through training. While we found exploration beneficial early in training, disabling it halfway through led to improved performance.
>
> [1] Schwarzer, Max, et al. "Bigger, better, faster: Human-level atari with human-level efficiency." International Conference on Machine Learning. PMLR, 2023.
>
> [2] Machado, Marlos C., et al. "Revisiting the arcade learning environment: Evaluation protocols and open problems for general agents." Journal of Artificial Intelligence Research 61 (2018): 523-562.
>
> [3] Kapturowski, Steven, et al. "Human-level atari 200x faster." arXiv preprint arXiv:2209.07550 (2022).

---

### Official Review · Reviewer_bNh2 · 2025-03-18

**Overall Recommendation:** 4

**Summary:**

The paper presents BTR, which integrates several well-established techniques to achieve strong performance on Atari and Procgen with limited computational resources. Through detailed ablation studies, the authors demonstrate how each component contributes to their method, showcasing the trade-offs between computational efficiency and RL performance. While falling short of DreamerV3 and MEME, BTR significantly surpasses Rainbow DQN on the Atari-60 benchmark in under 12 hours of training for 200 million frames on a Desktop PC.

**Claims And Evidence:**

1. The authors state that BTR outperforms the state-of-the-art performance for non-recurrent RL. However, the distinction of *non-recurrent* alone is not particularly meaningful or impressive, as employing RNN encoders is typically a deliberate design choice rather than a limitation. Although, in principle, this is a valid claim, due to the stronger baselines employing RNNs by default, I don’t think it is very noteworthy.
2. The authors claim that BTR can solve modern games by demonstrating performance on three Wii games. However, these games do not present significantly greater challenges than typical Atari or ProcGen tasks. They feature limited discrete action spaces, dense rewards, and lack the complexity, variety, and stochasticity characteristic of modern games:
- *Mortal Kombat: Armageddon*, despite its 3D physics engine, barely utilizes the 3rd dimension and effectively functions as a 2D side-scroller.
- *Mario Kart Wii*, with only 4 discrete actions and repetitive lap-based gameplay, offers limited complexity and variety.
- *Super Mario Galaxy* is fully deterministic, allowing an agent to directly overfit to a single level.

    To better substantiate the claim that BTR can handle modern games, the authors should evaluate it on more open-ended, truly 3D environments [1, 2].


[1] Raad, Maria Abi, et al. "Scaling instructable agents across many simulated worlds." *arXiv preprint arXiv:2404.10179* (2024).

[2] Fan, Linxi, et al. "Minedojo: Building open-ended embodied agents with internet-scale knowledge." *Advances in Neural Information Processing Systems* 35 (2022): 18343-18362.

**Essential References Not Discussed:**

Prior works have achieved higher performance on ProcGen with similar or lower data budgets [1, 2, 3].

[1] Jesson, Andrew, and Yiding Jiang. "Improving Generalization on the ProcGen Benchmark with Simple Architectural Changes and Scale." *arXiv preprint arXiv:2410.10905* (2024).

[2] Cobbe, Karl W., et al. "Phasic policy gradient." *International Conference on Machine Learning*. PMLR, 2021.

[3] Hafner, Danijar, et al. "Mastering diverse domains through world models." *arXiv preprint arXiv:2301.04104* (2023).

**Experimental Designs Or Analyses:**

1. The component impact analysis is well executed, clearly showing how each part of BTR affects performance and training hours on Atari. The authors convincingly demonstrate how each design choice contributes not only to BTR's performance improvements but also to other metrics such as the percentage of dormant neurons, network weights, SRank, action gaps, and policy churn. Detailing the network architecture choices, loss components, and hyperparameters is particularly helpful. Explicitly discussing which attempts did not work, further adds transparency and makes the results more credible.
2. The authors selectively omit evaluations of other baselines from key results without justification, artificially inflating the apparent superiority of BTR. **DreamerV3** and **MEME** achieve higher IQM scores on Atari-60 (9.6 vs. BTR’s 7.4, Table 3), yet they are excluded from Figures 1 and 2, Tables A1 and A2. Meanwhile, **PQN**, which runs ~20x faster but performs worse (Table A3), is omitted from Table 3, where BTR's lower walltime and network size are emphasized. This cherry-picking distorts the comparison and inflates BTR’s perceived advantage.
3. While demonstrating BTR on three Wii games demonstrates the algorithm's versatility, these games are not established benchmarks, making it difficult to determine how notable BTR’s performance is relative to other methods. Without baseline comparisons or prior evaluations, it is unclear whether BTR’s results represent a unique achievement or if comparable performance could be attained by existing methods. To strengthen the evaluation, the authors should include comparisons with other RL baselines. Notably, basic RL algorithms have managed to beat similar games such as *Super Mario Kart*, *Mortal Kombat 3*, and *Super Mario 64*, although in informal, non-academic contexts, such as blog posts and YouTube videos.

**Methods And Evaluation Criteria:**

Although the authors did not introduce novel algorithmic enhancements, their research is valuable as it effectively combines existing RL components into a cohesive and efficient algorithm. By integrating several independently validated improvements, the authors successfully demonstrate a method that achieves high performance while remaining computationally accessible, targeting smaller research labs and hobbyists with limited computing resources. The authors attempt to bridge the gap between low-cost simple algorithms and resource-intensive, cutting-edge RL methods, thoroughly justifying their design choices for the trade-offs in computational efficiency versus performance.

**Other Comments Or Suggestions:**

1. Section E.3 consists only of Tables E7 and E8, which are placed on the following page, making it difficult to follow. Explicit in-text references to these tables would improve clarity and readability.
2. For the camera-ready version, the authors could update the BTR results with LayerNorm, as Appendix H suggests it further improves performance.
3. Figures 2 and B2 include results for the baseline *REM*, but this method is never introduced or referenced in the paper. This baseline likely refers to Random Ensemble Mixture [1],
4. The authors could evaluate BTR on well-established image-based 3D benchmarks with discrete action spaces like DMLab [2] and ViZDoom [3].
5. Typos
    1. Lines 131-134 missing punctuation: *the convolutional layers,* ***which***
    2. Line 180 *Boo**t**strapping*
    3. Line 190 *with ~~with~~*
    4. Line 320 *we find **that** maxpooling*
    5. Line 1245 *many other ~~the other~~ techniques*
    6. Line 1376 *we use a forked repository ~~of~~ to allow*
    7. Mortal Combat —> Mortal **K**ombat

[1] Agarwal, Rishabh, Dale Schuurmans, and Mohammad Norouzi. "An optimistic perspective on offline reinforcement learning." *International conference on machine learning*. PMLR, 2020.

[2] Beattie, Charles, et al. "Deepmind lab." *arXiv preprint arXiv:1612.03801* (2016).

[3] Kempka, Michał, et al. "Vizdoom: A doom-based ai research platform for visual reinforcement learning." *2016 IEEE conference on computational intelligence and games (CIG)*. IEEE, 2016.

**Other Strengths And Weaknesses:**

I’ve outlined the strengths and weaknesses above.

**Questions For Authors:**

1. How is *consistent completion* established in the Wii games (Figure 3)?
2. Why aren’t the Atari results of MEME and DreamerV3 not included in Figure 1 and Figure 2?
3. How do other baselines perform on The Wii games?
4. What is noteworthy about non-recurrent RL?

**Relation To Broader Scientific Literature:**

The contributions of this paper clearly build on existing RL literature. The authors integrate 6 previously established methods and algorithmic tricks to work with Rainbow DQN to achieve strong performance with limited computational resources.

**Theoretical Claims:**

This paper does not contain theoretical claims or formal proofs; its primary contributions are experimental and methodological.

---

> ### Author Rebuttal · Authors · 2025-03-28
>
> We thank the reviewer for their very thorough and constructive review, which clearly took a great deal of time and effort. We appreciate that the reviewer clearly understands the value of producing an accessible and high-performance algorithm.
>
> **Complexity of Wii Games** - We would like to argue that in particular, Mario Kart Wii is a very challenging environment, more so than it has been given credit for. This environment contains 11 other racers and the use of randomized items, introducing a very high degree of stochasticity into the environment, far beyond the stochasticity introduced in Atari. Furthermore, we tested the agent on the track “Rainbow Road”, commonly known as the game’s hardest track, taking over a minute even for a single lap, and featuring a very complex and noisy observation space (fully 3D, with multicoloured roads and constantly changing 3D backgrounds that resemble the track). Although Super Mario Galaxy uses a single fixed level, we would argue this is unique and more challenging than typical benchmarks due to complex movement mechanics and high-resolution images.
>
> **Claim of SoTa for non-recurrent RL** - While non-recurrent RL may not be a category of interest, we chose this phrasing as it is indicative of walltime efficiency, since off-policy recurrent algorithms (particularly image-based ones) are very expensive [1]. We thought this better than stating “SoTa for walltime efficiency”, or “SoTa for Desktop PCs”, as these are somewhat tricky to formally verify (though we believe it is important to the field). As you mention, strong baselines heavily rely on expensive recurrent models.
>
> **Dreamer v3 and MEME baselines** - As you mention, we already included these baselines in Table 3; we did not include these results in Figures 1 and 2  as it would be misleading to compare BTR against algorithms which are inaccessible to our targeted use case (academics, hobbyists, etc). However, to be more transparent about BTR’s place among these algorithms, we would be happy to add Dreamer v3 and MEME to Figure 2 (the box plot), while also stating the walltime required for these algorithms to emphasize the difference. As for Figure 1, we can replace this with a walltime efficiency curve that includes Dreamer v3 as we feel this better aligns with the paper’s contributions (MEME, however, did not release sample efficiency curves to our knowledge). We didn’t include PQN in any main paper figures/tables as they used Life Information, which we discuss at length in Appendix I, and leads to completely non-comparable results and is not standard (Table 3 also uses Atari-60, not Atari-5, and we didn’t perform experiments on Atari-60 using life information).
>
> **Wii Game Results** – As you mention, the setting is somewhat informal, and thus we find it difficult to make direct comparisons to other benchmarks. We did not intend to use these environments as formal benchmarks (we already rely on Atari and Procgen for that purpose), but rather as a demonstration of what BTR is capable of. Furthermore, we have been unable to find any other work on Super Mario 64 that uses a reinforcement learning approach, and we would be interested if you could reference such work. As for Super Mario Kart, we would like to highlight the difference in complexity between it and Mario Kart Wii, as the latter includes far more complex graphics, tracks, and driving mechanics (e.g., mini-turbos, wheelies, tricks, etc.).
>
>
> **Missing References** – We are happy to add the references you suggested to our section on Procgen to provide a more comprehensive discussion. We also found some of the work you referenced [2] very interesting, and it may help improve BTR’s performance on the Procgen benchmark.
>
> **Comments and Suggestions** – We are happy to address the minor adjustments you suggested in points 1, 3, and 5 of this section. As for point 2, we also considered adding LayerNorm, though it may introduce inconsistencies across the ablations and other figures, and could complicate the narrative of the paper. Regarding point 4, we are currently running additional experiments for the VizDoom environment, which we will add to the appendix.
>
> **Questions** – We define consistent completion as achieving over a 90% success rate, and we will add this definition to the caption of Figure 3. We believe your remaining questions have been addressed throughout our response.
>
> [1] Kapturowski, Steven, et al. "Recurrent experience replay in distributed reinforcement learning." International conference on learning representations. 2018.
> [2] Jesson, Andrew, and Yiding Jiang. "Improving Generalization on the ProcGen Benchmark with Simple Architectural Changes and Scale." arXiv preprint arXiv:2410.10905 (2024).

---

> > ### Comment · Reviewer_bNh2 · 2025-04-04
> >
> > 1. **Wii Games**. I agree that the Wii games are more complex than most atari or procgen tasks, however, I am not sure to what extent. The environments’ more advanced graphics and physics certainly contribute to that, however RL agents tend to struggle more with things like sparse and long-horizon rewards. In Mario Kart, the agent can neglect 1) obtaining or using items, 2) other racer types, and 3) avoiding items being used on itself by others and still do a reasonable job beating its opponents. I doubt that the original high-resolution rendering in Super Mario Galaxy complicates the task. The agent is likely to learn an equally good policy from down-scaled inputs. It only needs to detect itself in relation to the platforms, walls, and obstacles. Nevertheless, I don't think this is a notable weakness of the paper, and the extra environments highlight BTR as a generally capable algorithm. If the authors wish to better demonstrate the complexity of these environments, I suggest evaluating other baselines on them.
> > 2. **SoTa RL**. Thanks for clarifying this. I understand it is difficult to find or justify a sweet-spot in the trade-off between performance and computational efficiency (as with any multi-objective problem with a trade-off). Although, the RSSM in Dreamer and MEME adds a lot of overhead because it cannot be parallelized, I don’t think it’s the main or only reason slowing down these models. It’s rather a combination of large, overparameterized networks, BPTT over full sequences, and running thousands of imagined rollouts in the latent space per update. Nevertheless, I cannot suggest a better aspect to distinguish BTR from the rest.
> > 3. **Stronger Baselines**. Figures 1 and 2 would certainly benefit from more recent baselines. The authors can themselves determine what is best to represent the results comparison, as long as it is reflective of stronger baselines, while highlighting the strengths of BTR in good nature. Regarding PQN, since including life information to BTR seemed like as easy adaptation, and since PQN runs very fast, why didn’t the authors run PQN themselves without the life information and on 200M (adapted to their setting), instead of reporting the results form the paper?
> >
> > Due to a convincing rebuttal of other reviews and mine, I have decided to increase the score. I believe BTR is a valuable contribution for low-budget setups, and that wallclock time is an important measure to enable a tight feedback loop for rapid experimentation with reasonable performance from a generally capable algorithm. The core weakness of the paper still, as also pointed out by other reviewers, is the omission of strong baselines from 1) recent literature targeting sample-efficiency (SR-SPR, EfficientZero, and BBF), and 2) Figures 1 and 2 without explanation. I suggest the authors incorporate these points in their final revision.

---

> > > ### Author Response · Authors · 2025-04-04
> > >
> > > We would like to thank the reviewer for their insightful discussion and raising of their score. We appreciate your suggestions and will consider these in future work.

---

### Decision · Program_Chairs · 2025-05-01

**Decision:**

Accept (poster)

**Comment:**

This paper presents BTR, an approach that integrates several well-established techniques to achieve strong performance on Atari and Procgen with limited computational resources. Although the algorithmic novelty is limited, the paper provides value by effectively combining existing reinforcement learning components into a cohesive and efficient framework. Additionally, the authors addressed the reviewers’ concerns regarding baselines thoroughly during the rebuttal. Therefore, I recommend acceptance.